**eLife** RESEARCH ARTICLE

# Coordinated *Tbx3/Tbx5* transcriptional control of the adult ventricular conduction system

Ozanna Burnicka-Turek[1]*, Katy A Trampel[2], Brigitte Laforest[1], Michael T Broman[3], Xinan H Yang[1], Zoheb Khan[1], Eric Rytkin[2], Binjie Li[2], Ella Schaffer[1], Margaret Gadek[1], Kaitlyn M Shen[1], Igor R Efimov[2], Ivan P Moskowitz[1]*

[1]Departments of Pediatrics, Pathology, and Human Genetics, University of Chicago, Chicago, United States; [2]Departments of Biomedical Engineering, Northwestern University, Chicago, United States; [3]Department of Medicine, Section of Cardiology, University of Chicago, Chicago, United States

## eLife Assessment

The work presented is **important** for our understanding of the development of the cardiac conduction system and its regulation by T-box transcription factors. The conclusions are supported by **convincing** data. Overall this is an excellent study that advances our understanding of cardiac biology and has implications beyond the immediate field of study.

**\*For correspondence:**
burnickatureko@uchicago.edu (OB-T);
imoskowitz@peds.bsd.uchicago.edu (IPM)

**Competing interest:** The authors declare that no competing interests exist.

**Abstract** The cardiac conduction system (CCS) orchestrates the electrical impulses that enable coordinated contraction of the cardiac chambers. The T-box transcription factors *TBX3* and *TBX5* are required for CCS development and associated with overlapping and distinct human CCS diseases. We evaluated the coordinated role of *Tbx3* and *Tbx5* in the murine ventricular conduction system (VCS). We engineered a compound *Tbx3:Tbx5* conditional knockout allele for both genes located in cis on mouse chromosome 5. Conditional deletion of both T-box transcriptional factors in the VCS, using the VCS-specific *MinK*^CreERT2, caused loss of VCS function and molecular identity. Combined *Tbx3* and *Tbx5* deficiency in the adult VCS led to conduction defects, including prolonged PR and QRS intervals and elevated susceptibility to ventricular tachycardia. These electrophysiological defects occurred prior to detectable alterations in cardiac contractility or histologic morphology, indicative of a primary conduction system defect. *Tbx3:Tbx5* double-knockout VCS cardiomyocytes revealed a transcriptional shift toward non-CCS-specialized working myocardium, indicating a change to their cellular identity. Furthermore, optical mapping revealed a loss of VCS-specific conduction system propagation. Collectively, these findings indicate that *Tbx3* and *Tbx5* coordinate to control VCS molecular fate and function, with implications for understanding cardiac conduction disorders in humans.

## Introduction

The cardiac conduction system (CCS) constitutes a highly specialized network of cardiomyocytes that initiate and propagate the electrical impulses required for synchronized contractions of the heart. In the mature mammalian heart, the functional components of the CCS can be broadly divided into the slowly propagating atrial nodes (~5 cm/s), containing the sinoatrial node (SAN) and atrioventricular node (AVN), and the rapidly propagating ventricular conduction system (VCS) (~200 cm/s), including the AV (His) bundle and the right and left bundle branches (BBs). The VCS

is responsible for rapid propagation of the electrical impulse from the AVN to the ventricular apex to enable synchronous ventricular contraction and effective ejection of blood from the ventricles (*Arnolds et al., 2012*; *Park and Fishman, 2011*; *Moskowitz et al., 2007*). Defects of CCS can occur in normally formed hearts as well as in patients with structural congenital heart disease and are a major source of morbidity and mortality (*Arnolds et al., 2012*; *Moskowitz et al., 2007*; *Munshi, 2012*; *Rubart and Zipes, 2005*; *Huikuri et al., 2001*). The VCS specifically has been recognized as a substrate for life-threatening ventricular arrhythmias, including bundle branch reentry tachycardia, idiopathic fascicular tachycardia, short-coupled torsade de pointes, and ventricular fibrillation (*Arnolds et al., 2012*; *Huikuri et al., 2001*; *van Duijvenboden et al., 2014*; *Arnolds and Moskowitz, 2011b*; *Scheinman, 2009*). Despite the severe clinical consequences of CCS disorders, the molecular mechanisms that establish and maintain regional functionality of the mature CCS domains require further study.

Human genetic studies have identified numerous loci associated with adult human CCS function, including the developmentally important factors *Tbx3* and *Tbx5* (reviewed in *Arnolds et al., 2012*; *Arnolds et al., 2011a*). *Tbx3* and *Tbx5* play crucial roles in adult CCS development and function (*Arnolds et al., 2012*; *Moskowitz et al., 2007*; *Burnicka-Turek et al., 2020*; *Moskowitz et al., 2004*; *van Weerd and Christoffels, 2016*; *van den Boogaard et al., 2012*; *Bakker et al., 2012*; *Hatcher and Basson, 2009*; *Bakker et al., 2008*; *Hoogaars et al., 2007b*; *Mori et al., 2006*; *Bruneau et al., 2001*). *Tbx5* encodes a T-box transcriptional activator required for structural and conduction system cardiac development (*Moskowitz et al., 2007*; *Moskowitz et al., 2004*; *Bruneau et al., 2001*; *Hoogaars et al., 2007a*). Dominant mutations in human *TBX5* cause Holt–Oram syndrome (HOS, OMIM:142900), an autosomal dominant disorder characterized by upper limb malformations, congenital heart defects, and CCS abnormalities (*Basson et al., 1997*; *Li et al., 1997*; *Basson et al., 1994*). The cardiac phenotype of HOS, including atrioventricular conduction delay, has been recapitulated in the *Tbx5* heterozygous mice (*Bruneau et al., 2001*). Moreover, VCS-specific *Tbx5* knockout caused slowed VCS function and ventricular tachycardia (VT) resulting in sudden death in mice (*Arnolds et al., 2012*), emphasizing the importance of *Tbx5* in VCS conduction. *Tbx5* is strongly expressed in the atria and VCS (*Arnolds et al., 2012*; *Moskowitz et al., 2007*; *Moskowitz et al., 2004*; *Bakker et al., 2008*), and directly regulates several targets required for VCS function (*Moskowitz et al., 2007*; *Bruneau et al., 2001*; *Hiroi et al., 2001*), including *Gja5 (Cx40)* (*Bruneau et al., 2001*) and *Scn5a (Nav1.5)* (*Arnolds et al., 2012*). *Tbx3* encodes a T-box transcriptional repressor which is critical for cardiac development (*Bakker et al., 2008*; *Frank et al., 2011*). Dominant mutations in human *TBX3* cause Ulnar–Mammary syndrome (OMIM:181450), a developmental disorder (*Meneghini et al., 2006*; *Bamshad et al., 1997*), that includes functional conduction system defects (*Linden et al., 2009*). In the heart, *Tbx3* is specifically expressed within CCS (*Bakker et al., 2008*; *Hoogaars et al., 2004*), and its deficiency below critical level leads to lethal arrhythmias (*Frank et al., 2011*). Furthermore, *Tbx3* is required for the molecular identity but not the function of the VCS (*Bakker et al., 2008*). In contrast, *Tbx3* in SAN and AVN is required for their proper function (*Hoogaars et al., 2004*), emphasizing its critical role in maintaining proper cardiac rhythm.

A model for regional CCS specialization suggests that the adult CCS is organized entirely as a slow conduction system ground state by *Tbx3* with a T-box-dependent, physiologically dominant fast conduction system network driven specifically in the VCS by *Tbx5* (*Burnicka-Turek et al., 2020*). The adult VCS-specific removal of TBX5 or overexpression of TBX3 shifted the fast VCS into a slow nodal-like system, indicating that the *Tbx3/Tbx5* ratio determines nodal versus VCS function (*Burnicka-Turek et al., 2020*). However, a comprehensive assessment of the coordinated requirements for *Tbx3* and *Tbx5* has been hindered by the inability to achieve their compound deletion due to their genomic proximity. *Tbx3* and *Tbx5* are situated in cis within 0.6 Mb on chromosome 5 in mice, rendering their simultaneous deletion unattainable with the available single allele conditional alleles. To investigate the consequences of *Tbx3* and *Tbx5* compound removal from the mature VCS, we generated a novel compound *Tbx3:Tbx5* double-conditional allele. We found that VCS-specific genetic removal of both TBX3 and TBX5 transformed fast-conducting, adult VCS into working myocardium-like cardiomyocytes, shifting them from conduction to non-conduction myocytes. These results demonstrated the coordinated requirements of both *Tbx3* and *Tbx5* for maintained specification of the mature VCS.

## Results

We generated a novel *Tbx3:Tbx5* double-floxed allele to enable the simultaneous conditional deletion of *Tbx3* and *Tbx5* genes specifically from the adult VCS. *Tbx3* and *Tbx5* reside in cis on mouse chromosome 5 (*Tbx3* mm39 chr5:119808734–119822789; *Tbx5* mm39 chr5:119970733–120023284). Therefore, to generate a double-conditional knockout, we targeted *Tbx5* in the background of a previously validated *Tbx3* floxed allele (*Frank et al., 2011*) using the CRISPR–Cas9 system (*Gurumurthy et al., 2021*; *Dow et al., 2015*; *Figure 1A*). We engineered a *Tbx5* floxed allele mirroring a previously published allele (*Bruneau et al., 2001*). This design enabled us to utilize the previously published individual *Tbx3* floxed allele (*Frank et al., 2011*) and individual *Tbx5* floxed allele (*Bruneau et al., 2001*) to serve as controls (*Figure 1A*, *Figure 1—figure supplements 1 and 2*).

We assessed the impact of removing *Tbx3* and *Tbx5* from the mature VCS by combining the *Tbx3:Tbx5* double-floxed mouse line (*Tbx3$^{fl/fl}$;Tbx5$^{fl/fl}$*) with a VCS-specific, tamoxifen (TM)-inducible *Cre* transgenic mouse line, *Kcne1$^{CreERT2}$* [Tg(RP23-276I20-MinKCreERT2)] – hereafter referred to as *MinK$^{CreERT2}$*, in accordance with existing literature (*Figure 1A*; *Arnolds et al., 2012*; *Arnolds and Moskowitz, 2011b*; *Burnicka-Turek et al., 2020*). Individual *Tbx3* floxed and *Tbx5* floxed mouse lines combined with *MinK$^{CreERT2}$* transgenic mouse lines (*Tbx3$^{fl/fl}$;Tbx5$^{+/+}$; R26R$^{eYFP/+}$;MinK$^{CreERT2/+}$* and *Tbx3$^{+/+}$;Tbx5$^{fl/fl}$;R26R$^{eYFP/+}$;MinK$^{CreERT2/+}$*, respectively) were generated as controls, and all allelic combinations were evaluated in a mixed genetic background. We compared VCS-specific *Tbx3:Tbx5* double-conditional mutants (*Tbx3$^{fl/fl}$;Tbx5$^{fl/fl}$;R26R$^{eYFP/+}$;MinK$^{CreERT2/+}$*) with control littermates (*Tbx3$^{+/+}$;Tbx5$^{+/+}$; R26R$^{eYFP/+}$;MinK$^{CreERT2/+}$*) and VCS-specific *Tbx3:Tbx5* double-conditional heterozygous littermates (*Tbx3$^{fl/+}$;Tbx5$^{fl/+}$;R26R$^{eYFP/+}$;MinK$^{CreERT2/+}$*). Additionally, we validated the newly created *Tbx5* floxed allele demonstrating that it is efficiently converted to the *Tbx5* null allele through *Cre* recombinase, causing a phenotype consistent with that observed from conversion of the previously published *Tbx5* floxed allele (*Arnolds et al., 2012*; *Bruneau et al., 2001*; *Figure 1—figure supplements 1 and 2*, Methods section).

We assessed experimental mice at 8–9 weeks of age following tamoxifen administration at 6 weeks of age (*Figure 1A*, Methods section). We observed loss of both *Tbx3* and *Tbx5* expression, on both the mRNA and protein levels, in the VCS of adult *Tbx3:Tbx5* double-conditional mutant mice (*Tbx3$^{fl/fl}$;Tbx5$^{fl/fl}$;R26R$^{eYFP/+}$;MinK$^{CreERT2/+}$*) but not in their littermate controls (*Tbx3$^{+/+}$;Tbx5$^{+/+}$;R26R$^{eYFP/+}$;MinK$^{CreERT2/+}$*) (*Figure 1B, C*). Partial loss of *Tbx3* and *Tbx5* expression in the adult VCS of *Tbx3:Tbx5* double-conditional heterozygous mice (*Tbx3$^{fl/+}$;Tbx5$^{fl/+}$;R26R$^{eYFP/+}$;MinK$^{CreERT2/+}$*) was observed compared to littermate controls (*Tbx3$^{+/+}$;Tbx5$^{+/+}$;R26R$^{eYFP/+}$;MinK$^{CreERT2/+}$*) (*Figure 1C*). We confirmed the specificity of the *Tbx3:Tbx5* double knockout for the VCS by assessing *Tbx3* and *Tbx5* expression levels in the atria and ventricles of tamoxifen-treated experimental mice. Consistent with the VCS selectivity of *Cre* activity in the *MinK$^{CreERT2}$* mice (*Arnolds and Moskowitz, 2011b*), *Tbx3* and *Tbx5* expression remained similar in the atrial and ventricular myocardium of all allelic combinations, including *Tbx3:Tbx5* double-conditional knockout mice (*Tbx3$^{fl/fl}$;Tbx5$^{fl/fl}$;R26R$^{eYFP/+}$;MinK$^{CreERT2/+}$*) (*Figure 1C*).

*Tbx3:Tbx5* double-conditional knockout mice (*Tbx3$^{fl/fl}$;Tbx5$^{fl/fl}$;R26R$^{eYFP/+}$; MinK$^{CreERT2/+}$*) appeared morphologically and functionally normal and indistinguishable from control littermates (*Tbx3$^{+/+}$;Tbx5$^{+/+}$;R26R$^{eYFP/+}$;MinK$^{CreERT2/+}$*) at 2 weeks post-tamoxifen. However, longitudinal analysis demonstrated sudden death of VCS-specific double-knockout mice beginning 3 weeks post-tamoxifen administration (*Figure 1D*). Within the 3 months post-tamoxifen administration, all tamoxifen-treated *Tbx3$^{fl/fl}$;Tbx5$^{fl/fl}$; R26R$^{eYFP/+}$;MinK$^{CreERT2/+}$* mice had died suddenly (*n* = 40). In contrast, no mortality was observed among the tamoxifen-treated *Tbx3$^{+/+}$;Tbx5$^{+/+}$;R26R$^{eYFP/+}$;MinK$^{CreERT2/+}$* or *Tbx3$^{fl/+}$;Tbx5$^{fl/+}$;R26R$^{eYFP/+}$;MinK$^{CreERT2/+}$* littermates (each cohort *n* = 40) during this period (*Tbx3:Tbx5* double-conditional knockout mice vs. control mice p < 0.0001; *Tbx3:Tbx5* double-conditional knockout mice vs. *Tbx3:Tbx5* double-conditional heterozygous mice p < 0.0001, log-rank test; *Figure 1D*). These results revealed that double deletion of *Tbx3* and *Tbx5* from the adult VCS causes lethality beginning at 3 weeks post-tamoxifen.

The onset of mortality observed in VCS-specific *Tbx3:Tbx5*-deficient mice starting at 3 weeks post-tamoxifen prompted us to investigate the electrophysiologic consequences of VCS-specific *Tbx3:Tbx5* double-knockout at 2–3 weeks post-tamoxifen, prior to the onset of lethality (*Figure 2*). VCS-specific *Tbx3:Tbx5* deficiency caused profound conduction slowing in *Tbx3$^{fl/fl}$;Tbx5$^{fl/fl}$;R26R$^{eYFP/+}$;MinK$^{CreERT2/+}$* mice by ambulatory telemetry electrocardiography (ECG) analysis compared to *Tbx3:Tbx5* double-conditional heterozygous mice (*Tbx3$^{fl/+}$;Tbx5$^{fl/+}$;R26R$^{eYFP/+}$;MinK$^{CreERT2/+}$*) and

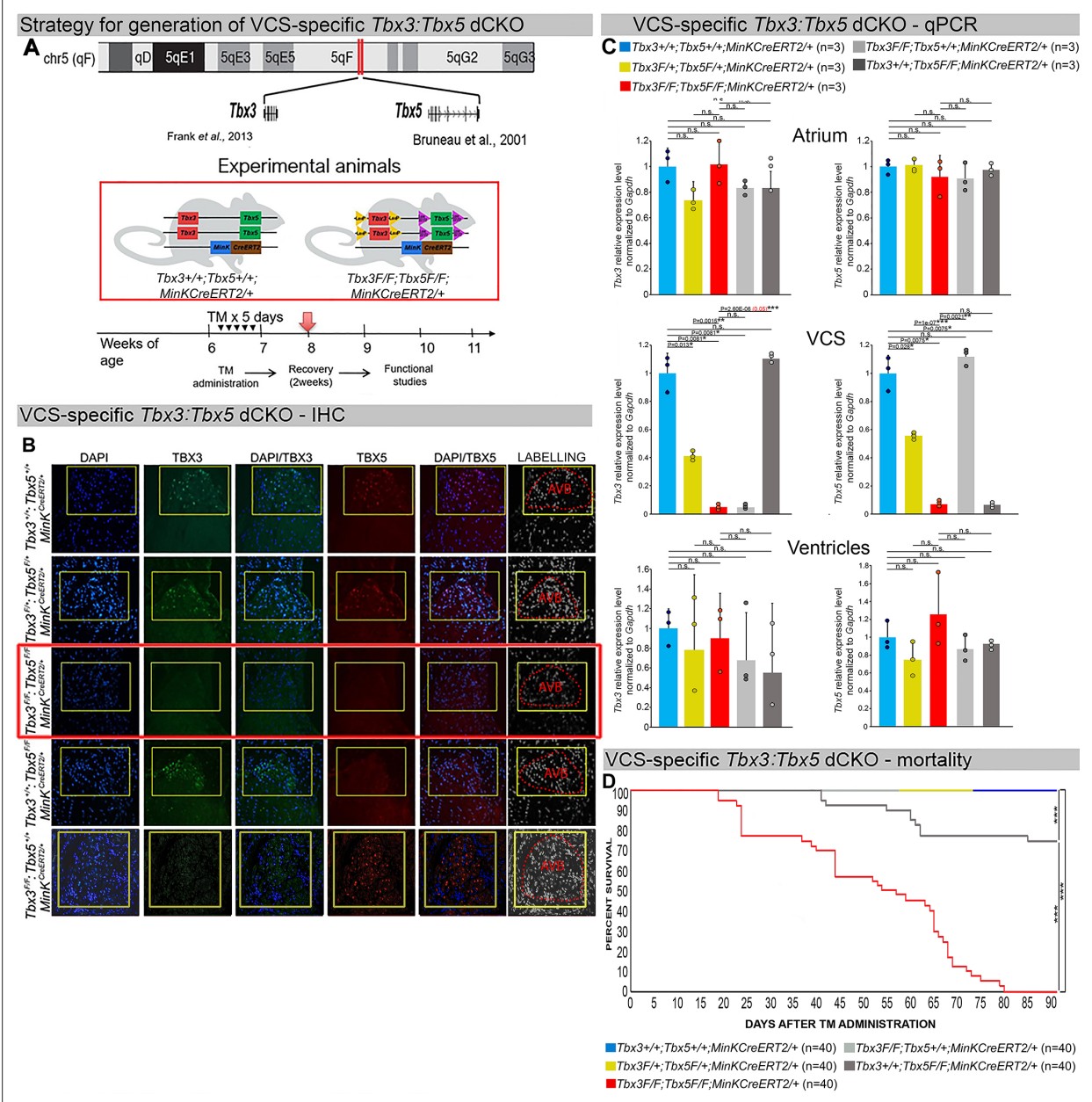

**Figure 1.** Generation of VCS-specific *Tbx3:Tbx5* double-conditional knockout mice. (**A**) Strategy to generate VCS-specific Tbx3:*Tbx5* double-conditional knockout mouse line. A new *Tbx3:Tbx5* double-conditional knockout mouse line (*Tbx3^fl/fl^;Tbx5^fl/fl^*) was generated using the CRISPR–Cas9 system (*Gurumurthy et al., 2021*; *Dow et al., 2015*) which allowed for the targeting of *Tbx5* in the background of the previously validated *Tbx3* floxed allele (*Frank et al., 2011*). A newly engineered *Tbx5* floxed allele has been developed to mirror a previously published allele (*Bruneau et al., 2001*). This design has enabled the utilization of the previously published individual *Tbx3* floxed allele (*Frank et al., 2011*) and individual *Tbx5* floxed allele (*Bruneau et al., 2001*) as controls. To conditionally delete *Tbx3* and *Tbx5* genes specifically from the adult VCS and generate the experimental animals, the *Tbx3:Tbx5* double-floxed mouse line (*Tbx3^fl/fl^;Tbx5^fl/fl^*) was combined with a VCS-specific tamoxifen inducible *Cre* transgenic mouse line (*MinK^CreERT2^* [Tg(RP23-276I20-*MinKCreERT2*) *Arnolds and Moskowitz, 2011b*]). All allelic combinations were generated and evaluated as littermates in a mixed genetic background. The experimental mice employed in all studies were administered tamoxifen at 6 weeks of age and subsequently evaluated at 9 weeks of age (3 weeks post-tamoxifen administration). The loss of *Tbx3* and *Tbx5* expression, on both the protein and mRNA levels, assessed by immunohistochemistry (**B**) and qRT-PCR (**C**), respectively, was observed in the VCS of adult *Tbx3:Tbx5* double-conditional mutant mice (*Tbx3^fl/fl^;Tbx5^fl/fl^;R26R^eYFP/+^;MinK^CreERT2/+^*), but not in their littermate controls (*Tbx3^+/+^;Tbx5^+/+^;R26R^eYFP/+^;MinK^CreERT2/+^*) (**B, C**). (**C**) qRT-PCR analysis showed a partial loss of *Tbx3* and *Tbx5* expression in the adult VCS of *Tbx3:Tbx5* double-conditional heterozygous mice (*Tbx3^fl/+^;Tbx5^fl/+^;R26R^eYFP/+^;MinK^CreERT2/+^*) compared to their littermate controls (*Tbx3^+/+^;Tbx5^+/+^;R26R^eYFP/+^;MinK^CreERT2/+^*). Additionally, qRT-PCR analysis confirmed the specificity of the *Tbx3:Tbx5* double knockout for the VCS by assessing *Tbx3* and *Tbx5* expression levels in the atria and ventricles of tamoxifen-treated experimental mice. Consistent with the VCS selectivity of *Cre* activity in the *MinK^CreERT2^* mice (*Arnolds and Moskowitz, 2011b*), *Tbx3* and *Tbx5* expression remained similar in the atrial and

*Figure 1 continued on next page*

*Figure 1 continued*

ventricular myocardium across all allelic combinations, including *Tbx3:Tbx5* double-conditional knockout mice (*Tbx3^{fl/fl};Tbx5^{fl/fl};R26R^{eYFP/+};MinK^{CreERT2/+}*). (**D**) Conducted longitudinal studies revealed a significantly increased mortality rate in VCS-specific *Tbx3:Tbx5*-deficient mice compared to their littermate controls (***p < 0.0001, log-rank test, GraphPad Prism), suggesting a requirement for both *Tbx3* and *Tbx5* in the mature VCS. All allelic combinations of experimental and control mice (n = 40 biological replicates/genotype) were followed longitudinally after tamoxifen administration at 6 weeks of age. *Tbx3:Tbx5* double-conditional knockout mice began to die suddenly at 3–4 weeks post-tamoxifen administration. Within the 3 months post-tamoxifen administration, all tamoxifen-treated *Tbx3^{fl/fl};Tbx5^{fl/fl};R26R^{eYFP/+};MinK^{CreERT2/+}* mice had died suddenly (n = 40) without previous signs of illness. In contrast, no mortality was observed among the tamoxifen-treated *Tbx3^{+/+};Tbx5^{+/+};R26R^{eYFP/+};MinK^{CreERT2/+}* and *Tbx3^{fl/+};Tbx5^{fl/+};R26R^{eYFP/+};MinK^{CreERT2/+}* littermates (each cohort n = 40) during this period. TBX3 and TBX5 protein expression was evaluated by immunohistochemistry (green and red signals, respectively) on serial sections from hearts of all allelic combinations (n = 3 biological replicates/genotype). Nuclei were stained with DAPI (blue signal). IHC original magnification: ×40. qRT-PCR data are presented as mean ± SD normalized to *Gapdh* and relative to *Tbx3^{+/+};Tbx5^{+/+};R26R^{eYFP/+};MinK^{CreERT2/+}* mice (set as 1). N = 3 biological replicates/genotype (VCS cardiomyocytes pooled from 30 mice per biological replicate); multiple testing correction using Benjamini and Hochberg procedure. Significance was assessed by Welch *t*-test (*FDR <0.05; **FDR <0.005; ***FDR <0.001) and confirmed by one-tailed Wilcoxon test (p value in parentheses) when normally distribution was rejected. Abbreviations: AVB, atrioventricular bundle (also known as bundle of His); FDR, false discovery rate; VCS, ventricular conduction system.

The online version of this article includes the following source data and figure supplement(s) for figure 1:

**Figure supplement 1.** Generation of a novel *Tbx5* floxed allele in a *Tbx3* floxed background.

**Figure supplement 1—source data 1.** File containing original gels corresponding to *Figure 1—figure supplement 1B*.

**Figure supplement 1—source data 2.** Zipped folder containing original gels corresponding to *Figure 1—figure supplement 1B*.

**Figure supplement 2.** Validation of the newly generated *Tbx5* floxed allele.

**Figure supplement 2—source data 1.** File containing original gels corresponding to *Figure 1—figure supplement 2B*.

**Figure supplement 2—source data 2.** Zipped folder containing original gels corresponding to *Figure 1—figure supplement 2B*.

**Figure supplement 3.** WT sequence of the targeted region at the mouse *Tbx5* locus.

**Figure supplement 4.** Sequence of long ssDNA donor designed to target exon 3 of the mouse *Tbx5* locus.

**Figure supplement 5.** Sequence of mouse new *Tbx5* floxed allele generated using CRISPR/Cas9 system with lssDNA composed of targeted exon 3 flanked by two lox2272 sites.

littermate controls (*Tbx3^{+/+};Tbx5^{+/+};R26R^{eYFP/+};MinK^{CreERT2/+}*) (*Figure 2A–F*). Specifically, the PR interval, representing the period between atrial and ventricular depolarization, and the QRS duration, indicating the length of ventricular depolarization and early repolarization in mice, were both significantly prolonged (*Figure 2A, B, D*; PR: *Tbx3:Tbx5* double-conditional knockout mice vs. control mice p < 0.05, n = 7, Welch *t*-test; QRS: *Tbx3:Tbx5* double-conditional knockout mice vs. control mice p < 0.05, n = 7, Welch *t*-test). Removal of both *Tbx3* and *Tbx5* from the adult VCS resulted in increased episodes of spontaneous VT. Ambulatory studies revealed episodes of spontaneous VT in four out of seven *Tbx3^{fl/fl};Tbx5^{fl/fl};R26R^{eYFP/+};MinK^{CreERT2/+}* mice, in contrast to none observed in seven littermate controls (*Figure 2G*, p < 0.05, n = 7, Welch *t*-test). Furthermore, *Tbx3:Tbx5* double-conditional knockout mice (*Tbx3^{fl/fl};Tbx5^{fl/fl};R26R^{eYFP/+};MinK^{CreERT2/+}*) showed significantly increased susceptibility to VT following burst stimulation in invasive electrophysiology (EP) studies (three of three *Tbx3^{fl/fl};Tbx5^{fl/fl};R26R^{eYFP/+};MinK^{CreERT2/+}* mice vs. zero of seven littermate controls; *Figure 2H*, p < 0.05, Welch *t*-test). In contrast, VCS-specific *Tbx3:Tbx5* double-conditional heterozygous mice (n = 7) showed neither conduction nor electrophysiological defects (*Figure 2*). Consistent with the use of a VCS-specific *Cre* (*MinK^{CreERT2}*), no changes in the refractory/recovery periods of atrium, ventricle, or nodes (atrial effective refractory period, ventricular effective refractory period (VERP), atrioventricular nodal effective refractory period, or sinus node recovery time) were detected by intracardiac EP conducted on experimental and control mice (*Figure 2I*, p < 0.05, Welch *t*-test).

To distinguish a primary conduction system abnormality from a secondary conduction abnormality resulting from cardiac dysfunction or remodeling, we evaluated cardiac form and function at the time of arrhythmia assessment, 2–3 weeks post-tamoxifen treatment (*Figure 3*). Transthoracic echocardiography revealed no significant differences in left ventricular ejection fraction (LVEF) and fractional shortening (FS) between VCS-specific *Tbx3:Tbx5*-deficient (*Tbx3^{fl/fl};Tbx5^{fl/fl};R26R^{eYFP/+};MinK^{CreERT2/+}*) and control (*Tbx3^{+/+};Tbx5^{+/+};R26R^{eYFP/+};MinK^{CreERT2/+}*) mice (*LVEF*: *Tbx3:Tbx5* double-conditional knockout mice vs. control mice p < 0.05, n = 7, Welch *t*-test; and FS: *Tbx3:Tbx5* double-conditional knockout mice vs. control mice p > 0.05, n = 7, Welch *t*-test; *Figure 3A, B, D*). Histological examination of all four chambers demonstrated no discernible differences between VCS-specific *Tbx3:Tbx5* double-knockout

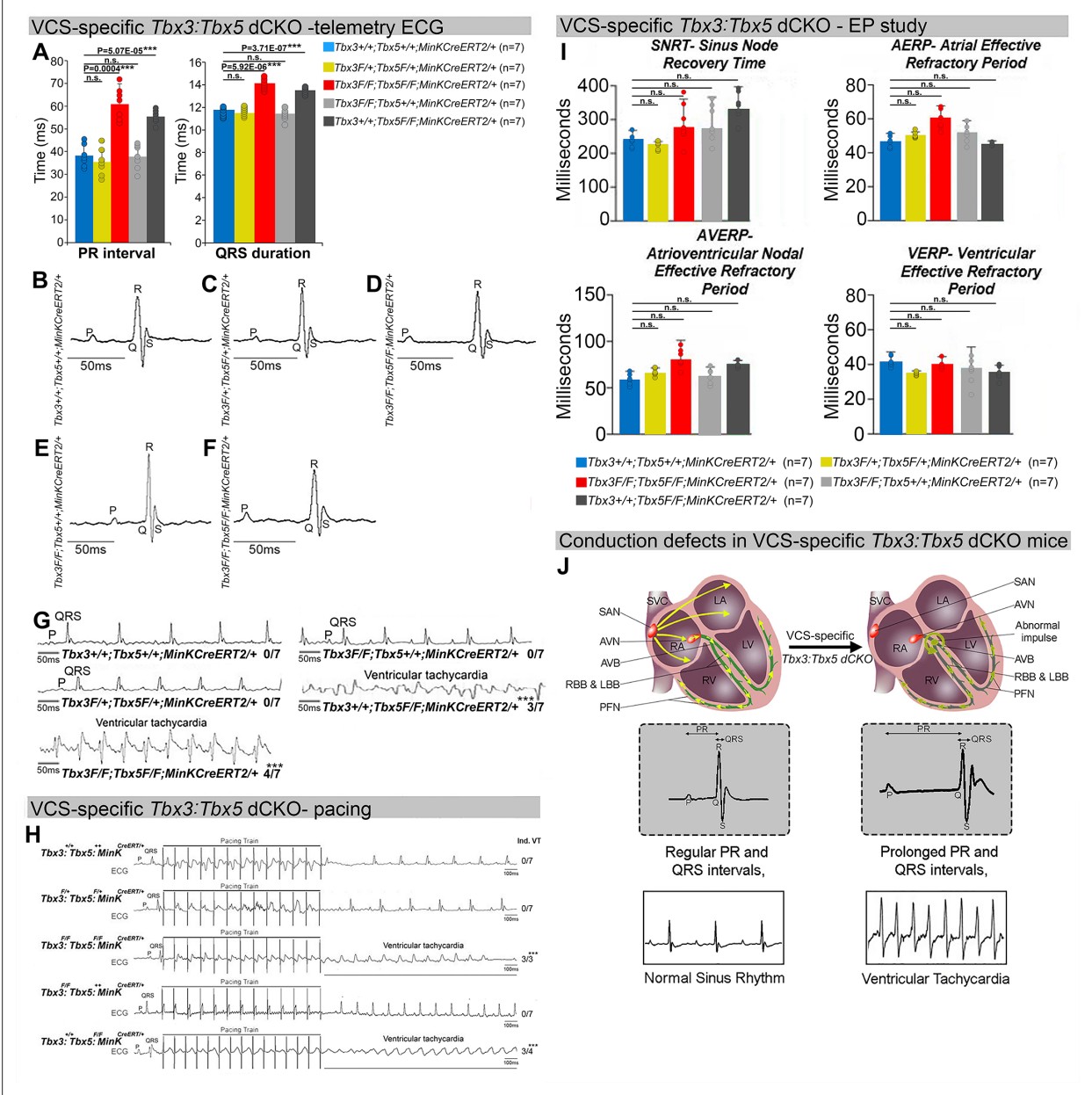

**Figure 2.** Arrhythmias and conduction abnormalities in mice with VCS-specific *Tbx3:Tbx5* double-conditional knockout. (**A–F**) VCS-specific *Tbx3:Tbx5* double-conditional knockout causes significant VCS conduction slowing in adult *Tbx3fl/fl;Tbx5fl/fl;R26ReYFP/+;MinKCreERT2/+* mice. (**A**) PR (left graph) and QRS (right graph) intervals calculated from ambulatory telemetry electrocardiography (ECG) recordings in (**B–F**). *Tbx3:Tbx5* double-conditional adult mice (*Tbx3fl/fl;Tbx5fl/fl;R26ReYFP/+;MinKCreERT2/+*) displayed significant PR and QRS intervals prolongation compared to littermate controls (*Tbx3+/+;Tbx5+/+;R26ReYFP/+;MinKCreERT2/+*) (A left and right graphs, respectively). Data are presented as mean ± SD. N = 7 biological replicates/genotype, multiple testing correction using Benjamini and Hochberg procedure; *Welch *t*-test p < 0.05 and FDR <0.05; **Welch *t*-test p < 0.005 and FDR <0.05; ***Welch *t*-test p < 0.001 and FDR <0.05. Representative ambulatory telemetry ECG of *Tbx3+/+;Tbx5+/+; R26ReYFP/+;MinKCreERT2/+* (**B**), *Tbx3fl/+;Tbx5fl/+;R26ReYFP/+;MinKCreERT2/+* (**C**), *Tbx3fl/fl;Tbx5fl/fl; R26ReYFP/+;MinKCreERT2/+* (**D**), *Tbx3fl/fl;Tbx5+/+;R26ReYFP/+;MinKCreERT2/+* (**E**), *Tbx3+/+;Tbx5fl/fl;R26ReYFP/+;MinKCreERT2/+* (**F**) mice. (**G**) Simultaneous genetic removal of *Tbx3* and *Tbx5* from the adult VCS resulted in significantly increased episodes of spontaneous ventricular tachycardia. Episodes of spontaneous ventricular tachycardia were observed in four of seven *Tbx3fl/fl;Tbx5fl/fl;R26ReYFP/+; MinKCreERT2/+* mice versus zero of seven littermate controls (*Tbx3+/+;Tbx5+/+;R26ReYFP/+; MinKCreERT2/+*) in ambulatory studies. N = 7 biological replicates/genotype; multiple testing correction using Benjamini and Hochberg procedure; *FDR of Welch *t*-test ≤0.05. (**H**) *Tbx3:Tbx5* double-conditional knockout mice (*Tbx3fl/fl;Tbx5fl/fl;R26ReYFP/+;MinKCreERT2/+*) showed significantly increased susceptibility to ventricular tachycardia following burst stimulation in invasive electrophysiology studies (three of three *Tbx3fl/fl;Tbx5fl/fl;R26ReYFP/+; MinKCreERT2/+* mice vs. zero of seven control *Tbx3+/+;Tbx5+/+;R26ReYFP/+;MinKCreERT2/+* mice. Fisher's exact test: *p < 0.05; n = 7 biological replicates/genotype. (**I**) Intracardiac electrophysiology detected no significant changes in SNRT, AERP, AVERP, and VERP recorded from experimental and control animals (n = 7 biological replicates/genotype; multiple testing correction using Benjamini

*Figure 2 continued on next page*

*Figure 2 continued*

and Hochberg procedure; *FDR of Welch *t*-test ≤0.05). (**J**) Graphical summary of conduction defects observed in adult, VCS-specific *Tbx3:Tbx5*-deficient mice. Simultaneous genetic deletion of *Tbx3* and *Tbx5* from the mature VCS results in conduction slowing, prolonged PR and QRS intervals, as well as ventricular tachycardia. Abbreviations: AVB, atrioventricular bundle (also known as bundle of His); AVN, atrioventricular node; FDR, false discovery rate; LA, left atrium; LBB, left bundle branches; LV, left ventricle; PFN, Purkinje fiber network; RA, right atrium; RBB, right bundle branches; RV, right ventricle; SAN, sinoatrial node; SVC, superior vena cava; VCS, ventricular conduction system.

($Tbx3^{fl/fl}$;$Tbx5^{fl/fl}$; $R26R^{eYFP/+}$;$MinK^{CreERT2/+}$) and control ($Tbx3^{+/+}$;$Tbx5^{+/+}$;$R26R^{eYFP/+}$; $MinK^{CreERT2/+}$) mice, nor between the double-knockout ($Tbx3^{fl/fl}$;$Tbx5^{fl/fl}$;$R26R^{eYFP/+}$; $MinK^{CreERT2/+}$) and single-knockout models for either *Tbx3* ($Tbx3^{fl/fl}$;$Tbx5^{+/+}$;$R26R^{eYFP/+}$;$MinK^{CreERT2/+}$) or *Tbx5* ($Tbx3^{+/+}$;$Tbx5^{fl/fl}$;$R26R^{eYFP/+}$;$MinK^{CreERT2/+}$). Ventricular muscle appeared normal without hypertrophy or myofibrillar disarray, and no fibrosis was present (*Figure 3G, I–K*, respectively). qRT-PCR analysis for fibrosis genes *Col1a1* (*Pan et al., 2022*; *Hua et al., 2020*; *Zhao et al., 2018*) and *Postn* (*Zhao et al., 2018*; *Ackerman et al., 2024*; *Wu et al., 2024*; *Oka et al., 2007*) further confirmed no fibrosis in VCS of *Tbx3:Tbx5*-deficient mice (*Figure 3L*). No contractile dysfunction, histological abnormalities, or increased expression of fibrosis genes were observed in VCS-specific *Tbx3:Tbx5* double-conditional heterozygous mice ($Tbx3^{fl/+}$;$Tbx5^{fl/+}$;$R26R^{eYFP/+}$; $MinK^{CreERT2/+}$) (*Figure 3A, C, H, L*). Taken together, these data indicate that the conduction defect and VT observed in mice with VCS-specific *Tbx3:Tbx5* deletion (*Figure 2*) occur prior to the onset of left ventricular (LV) dysfunction or evidence of remodeling (*Figure 3*), implying a primary origin.

To assess the hypothesis that *Tbx3* and *Tbx5* collectively promote VCS versus working myocardium phenotype, we conducted a transcriptional characterization of the adult VCS in $Tbx3^{fl/fl}$;$Tbx5^{fl/fl}$;$R26R^{eYFP/+}$;$MinK^{CreERT2/+}$ mutant mice compared to their $Tbx3^{+/+}$;$Tbx5^{+/+}$;$R26R^{eYFP/+}$;$MinK^{CreERT2/+}$ control littermates using three distinct sets of molecular markers by qRT-PCR (*Figure 4A–C*). The first set encompassed genes expressed throughout the entire conduction system (Pan-CCS) and implicated in slow-conducting nodal phenotype, such as *Hcn1*, *Hcn4*, *Cacna1d* (Cav1.3), *Cacna1g* (Cav3.1d), *Cacna1h* (Cav3.2), *Gjd3* (Cx30.2), and *Gjc1* (Cx45) (*Bakker et al., 2012*; *Hoogaars et al., 2004*; *van Eif et al., 2020*; *Verheule and Kaese, 2013*; *Liang et al., 2013*; *Greener et al., 2011*; *Mangoni et al., 2006*; *Marionneau et al., 2005*; *Garcia-Frigola et al., 2003*; *Schram et al., 2002*; *Figure 4A*). The second set included genes highly expressed in the fast-conducting VCS and important for VCS function, including *Gja5* (Cx40), *Scn5a* (Nav1.5), *Ryr2*, *Kcnk3* (Task-1), *Kcnj2* (Kir2.1), *Kcnj3* (Kir3.1), *Kcnj4* (IRK3), and *Kcnj12* (Kir2.2) (*Figure 4B*; *van Eif et al., 2020*; *Verheule and Kaese, 2013*; *Schram et al., 2002*; *Donner et al., 2011*; *Miquerol et al., 2010*; *Remme et al., 2009*; *Graham et al., 2006*). The third set contained markers specifically present in the working myocardium but absent in the CCS, such as *Gja1* (Cx43) and *Smpx* (*Figure 4C*; *Hoogaars et al., 2004*; *van Eif et al., 2020*; *Alcoléa et al., 1999*; *Kempen et al., 1996*). VCS-specific *Tbx3:Tbx5*-deficient mice lost VCS expression profile of genes required for the fast ventricular conduction (*Figure 4B*) as well as genes normally expressed in whole CCS (Pan-CCS genes) (*Figure 4A*). In contrast, these mice obtained VCS expression of working myocardium-specific molecular markers (*Figure 4C*). Immunoblotting analysis confirmed transcriptional changes observed by qRT-PCR (*Figure 4D*). This molecular characterization indicated that the *Tbx3:Tbx5* double mutant VCS adopted a gene expression profile similar to wild-type working myocardial-like cells (*Figure 4*).

The impact of the *Tbx3:Tbx5* double-conditional knockout on electrical impulse propagation in the VCS of the heart was assessed with optical mapping of the anterior epicardial surface of the ventricles and right septal preparations where VCS function should be observed (*Figure 5*). Optical mapping records changes in transmembrane potential from multiple cells in tissue preparations, where ventricular septal optical action potentials (OAPs) have two distinct action potential upstrokes. The first peak is a result of depolarization of the specialized fast VCS, followed by depolarization of ventricular working myocardium.

To visualize electrical impulse propagation in *Tbx3:Tbx5* double-conditional knockout mice, a 100 × 100 pixel data frame was plotted for the entire field of view of the right septal preparation (*Figure 5A, B*). For enhanced analysis of the VCS, the region encompassing the His bundle was distinguished in a red 10 × 10 area (*Figure 5B*). To specifically assess electrical impulse propagation within the His bundle, the same 10 × 10 pixel area representing this region was isolated from adult hearts of both control and *Tbx3:Tbx5* double-conditional knockout mice (*Figure 5C*). We observed only one OAP upstroke in *Tbx3:Tbx5* double-conditional knockout mice in contrast to control mice which showed two

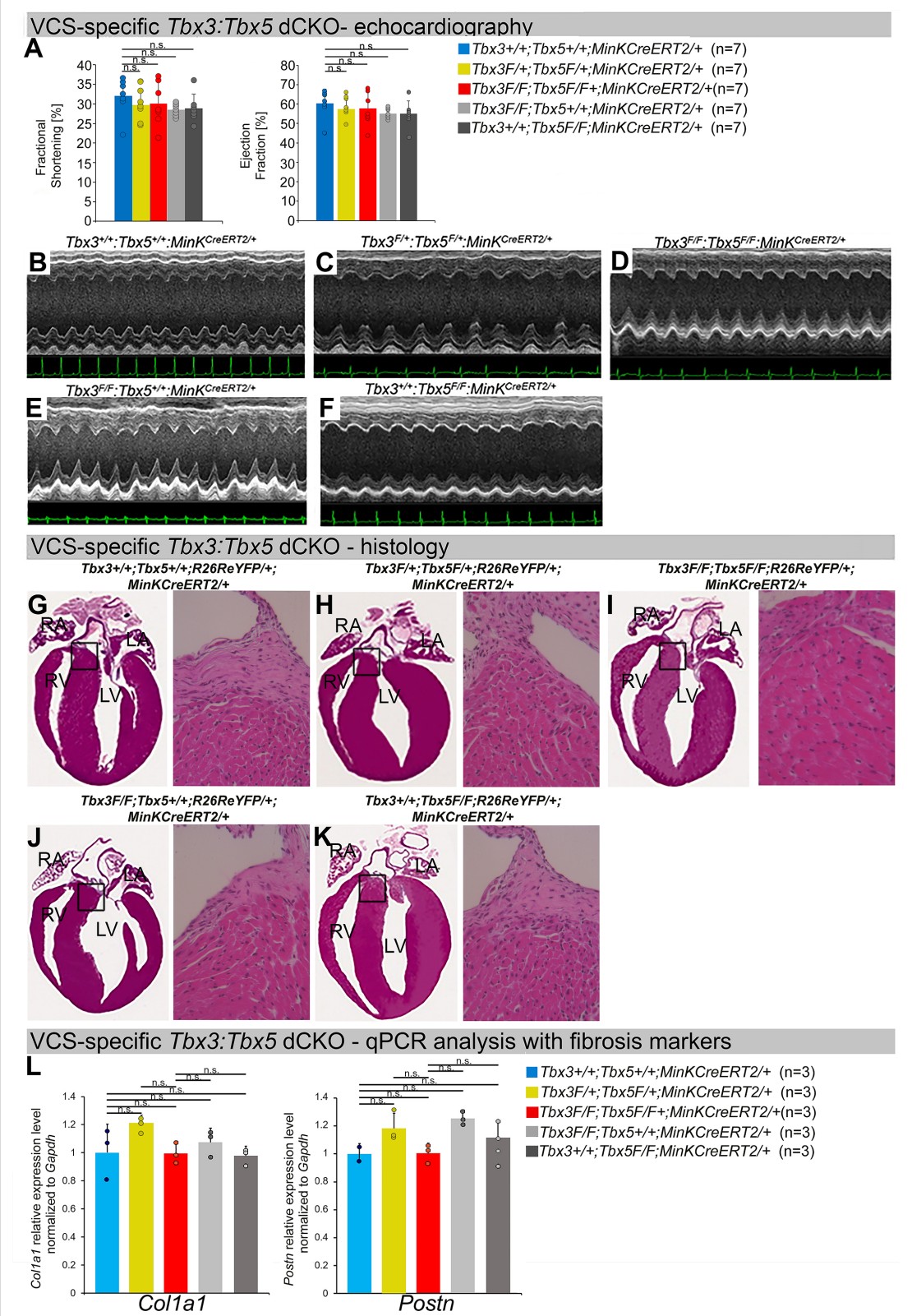

Figure 3. Cardiac function is preserved following double-conditional loss of *Tbx3* and *Tbx5* in the adult ventricular conduction system (VCS). (**A**) Left ventricular (LV) fractional shortening (**left graph**) and LV ejection fraction (**right graph**) calculated from the M-mode electrocardiographies (ECGs) in (**B–F**) revealed no contractile dysfunction in VCS-specific *Tbx3:Tbx5* double-conditional mutant mice (*Tbx3^{fl/fl};Tbx5^{fl/fl};R26R^{eYFP/+}; MinK^{CreERT2/+}*). Data are presented as mean ± SD. Welch *t*-test: ns, not significant (p > 0.05) ; *n* = 7 biological replicates/genotype. (**B–F**) Cardiac function, assessed by M-mode

*Figure 3 continued on next page*

Figure 3 continued

echocardiography from $Tbx3^{+/+};Tbx5^{+/+};R26R^{eYFP/+};MinK^{CreERT2/+}$ (B), $Tbx3^{fl/+};Tbx5^{fl/+};R26R^{eYFP/+};MinK^{CreERT2/+}$ (C), $Tbx3^{fl/fl};Tbx5^{fl/fl};R26R^{eYFP/+};MinK^{CreERT2/+}$ (D), $Tbx3^{fl/fl};Tbx5^{+/+};R26R^{eYFP/+};MinK^{CreERT2/+}$ (E), $Tbx3^{+/+};Tbx5^{fl/fl};R26R^{eYFP/+};MinK^{CreERT2/+}$ (F) mice shown above surface ECGs. No functional differences between mutant and control mice were detected. The most representative images for each genotype were utilized in the figure. $n = 7$ biological replicates/genotype. (G–K) Histological examination of all four chambers from $Tbx3^{+/+};Tbx5^{+/+};R26R^{eYFP/+};MinK^{CreERT2/+}$ (G), $Tbx3^{fl/+};Tbx5^{fl/+};R26R^{eYFP/+};MinK^{CreERT2/+}$ (H), $Tbx3^{fl/fl};Tbx5^{fl/fl};R26R^{eYFP/+};MinK^{CreERT2/+}$ (I), $Tbx3^{fl/fl};Tbx5^{+/+};R26R^{eYFP/+};MinK^{CreERT2/+}$ (J), $Tbx3^{+/+};Tbx5^{fl/fl};R26R^{eYFP/+};MinK^{CreERT2/+}$ (K) mice showed no histological abnormalities. The most representative images for each genotype were utilized in the figure. $n = 3–4$ biological replicates/genotype. Boxed areas in (G–K) have been shown at higher magnification at their right sides. (L) qRT-PCR analysis for fibrosis genes Col1a1 and Postn confirmed that there was no increase in expression of fibrosis markers in the VCS of Tbx3:Tbx5-deficient mice. Data are presented as mean ± SD normalized to Gapdh and relative to $Tbx3^{+/+};Tbx5^{+/+};R26R^{eYFP/+};MinK^{CreERT2/+}$ mice (set as 1). ns: FDR >0.05 in both Welch t-test and Wilcoxon test; $n = 2–3$ biological replicates/genotype (VCS cardiomyocytes pooled from 30 mice per biological replicate). Histological examination original magnification: ×2.5, boxed area showed at the higher magnification: ×40. Abbreviations: FDR, false discovery rate; LA, left atrium; RA, right atrium; LV, left ventricle; RV, right ventricle.

OAP upstrokes. The first derivative of the OAP (dOAP/dt) from the region of the His bundle was calculated and plotted in a dOAP/dt map further highlighting the number of temporally distinct upstrokes within the OAP (Figure 5D). In control mice, we observed two distinct depolarization upstrokes, one pertaining to the VCS and the second for working ventricular myocardium. However, in Tbx3:Tbx5 double-conditional knockout mice, only one maximum dOAP/dt was observed (Figure 5D). The OAP morphology observed in Tbx3:Tbx5 double-conditional knockout mice suggested a loss or reduction of the specialized fast VCS conduction (Figure 5C, D).

OAPs from the anterior epicardial surface of the ventricles were compared between control littermates and Tbx3:Tbx5 double-conditional knockout mice to assess changes in electrical impulse propagation in the ventricular working myocardium. Paced at a basic cycle length of 125 ms, control littermates and Tbx3:Tbx5 double-conditional knockout mice did not exhibit significant differences in action potential duration at 50% (APD50) or 80% (APD80) repolarization (Figure 5E, F). These observations suggested that the electrical activity of the ventricular working myocardium was not appreciably altered in double-knockout mice compared to controls. This finding is consistent with the lack of changes in the VERP observed in invasive EP studies of both control and Tbx3:Tbx5 double-conditional knockout mice (Figure 3I).

We further predicted that the double knockout would not affect action potentials or conduction properties distal to the VCS (Figure 6). OAP and dOAP/dt maps were created to observe the effects on electrical impulse propagation distal to the His bundle on the right septal preparation (Figure 6), similar to the methods applied to create OAP and dOAP/dt maps in the region of the His bundle (Figure 5), with the key difference being that the signals were recorded from the red 10 × 10 pixel regions plotted in the working ventricular myocardium distal to the His bundle instead of in the area of the His bundle (Figure 6A, B vs. Figure 5A, B, respectively). In both control littermates and Tbx3:Tbx5 double-conditional knockout mice, only one action potential upstroke and one dOAP/dt maximum are observed, which indicates that most of this region consists of the ventricular working myocardium (Figure 6C and D, respectively). In summary, the significant remodeling of electrical impulse propagation due to Tbx3:Tbx5 double-conditional knockout was observed by the loss of a distinct fast VCS impulse in the region of the His bundle, but not on the anterior epicardial surface of the ventricles or distal to the His bundle on the right ventricular (RV) septum (Figures 5 and 6).

## Discussion

Our study investigated the impact of compound Tbx3:Tbx5 deficiency on mature VCS function and molecular identity. Using a double-conditional knockout strategy, both genes were targeted specifically in the adult VCS. Loss of Tbx3 and Tbx5 expression in the mature VCS led to profound conduction defects, characterized by prolonged PR interval and QRS duration, increased susceptibility to VT, and sudden death. These alterations were observed in the absence of discernible changes in cardiac contractility or histological morphology, indicating a primary conduction system defect. Molecular characterization of the adult VCS unveiled an altered gene expression profile in Tbx3:Tbx5 double-conditional knockout mice, suggesting a transition from the distinctive fast VCS transcriptional profile to that resembling ventricular working myocardium. Optical mapping demonstrated loss of the specialized fast VCS function in Tbx3:Tbx5 double-conditional knockout mice, further suggesting that this region acquired an electrophysiological phenotype similar to the ventricular working myocardium.

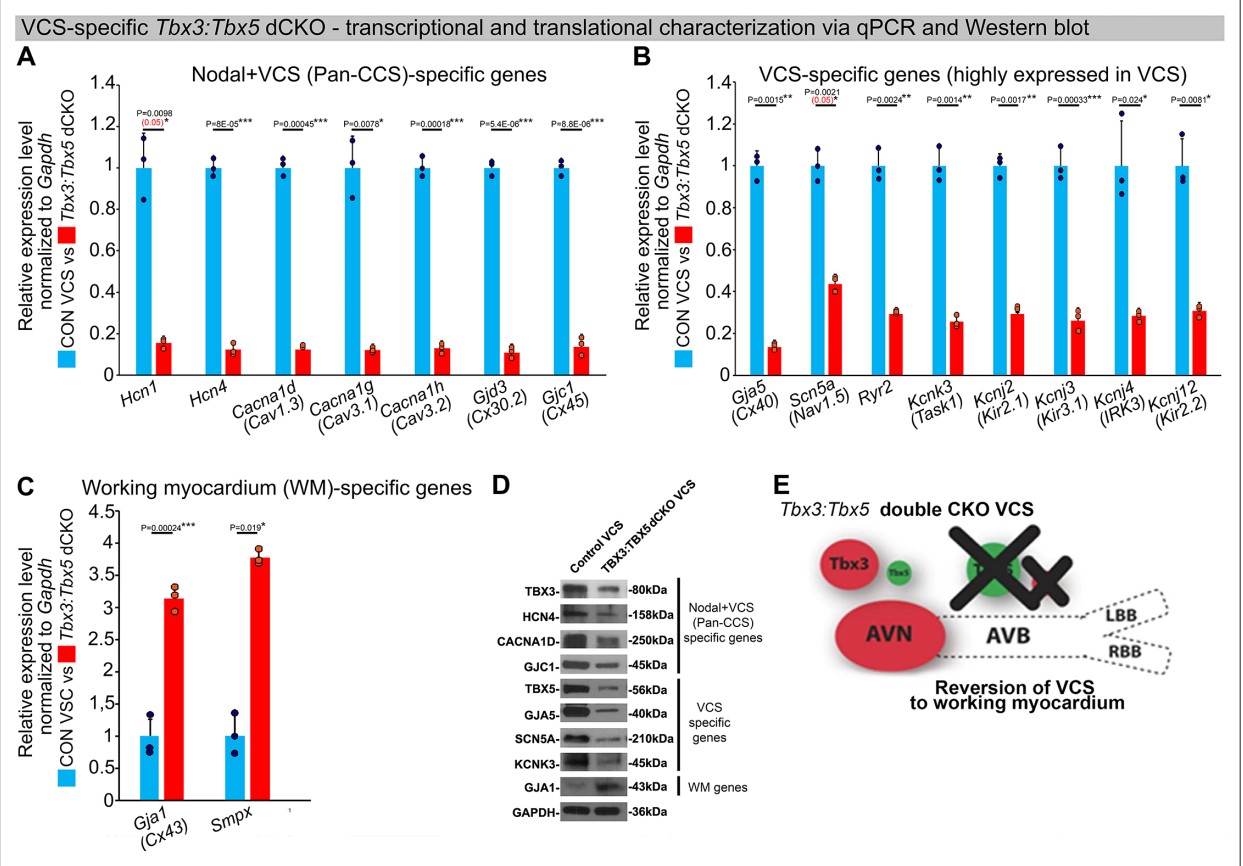

**Figure 4.** In the adult murine heart, *Tbx3* and *Tbx5* collectively promote ventricular conduction system (VCS) versus working myocardium (WM) phenotype. (**A–C**) qRT-PCR analysis of molecular changes driven by VCS-specific *Tbx3:Tbx5* double-conditional knockout in adult mice. Transcriptional characterization of the adult VCS in *Tbx3*$^{fl/fl}$;*Tbx5*$^{fl/fl}$;*R26R*$^{eYFP/+}$;*MinK*$^{CreERT2/+}$ mutant mice, compared to their *Tbx3*$^{+/+}$;*Tbx5*$^{+/+}$;*R26R*$^{eYFP/+}$;*MinK*$^{CreERT2/+}$ control littermates, was conducted using three distinct sets of molecular markers. (**A**) Genes expressed throughout the entire conduction system (Pan-CCS), implicated in the slow-conducting nodal phenotype. (**B**) Genes highly expressed in the fast-conducting VCS, critical for VCS function. (**C**) Markers specifically present in the working myocardium but absent in the CCS. VCS-specific *Tbx3:Tbx5*-deficient mice lost the VCS expression profile, including genes necessary for fast ventricular conduction (**B**) and those typically expressed in the entire CCS (Pan-CCS genes), essential for the slow-conducting nodal phenotype (**A**). In contrast, they acquired VCS expression of working myocardium-specific molecular markers important for working myocardial function (**C**). (**D**) Immunoblotting analysis confirmed transcriptional changes indicated by qRT-PCR analysis (**A–C**) in VCS-specific *Tbx3:Tbx5* double-conditional knockout in adult mice. (**E**) Graphical summary of transcriptional changes observed in VCS of VCS-specific *Tbx3:Tbx5*-deficient mice. Simultaneous genetic deletion of *Tbx3* and *Tbx5* from the mature VCS resulted in a transcriptional profile resembling that of ventricular working myocardium. qRT-PCR data are presented as mean ± SD normalized to *Gapdh* and relative to *Tbx3*$^{+/+}$;*Tbx5*$^{+/+}$;*R26R*$^{eYFP/+}$;*MinK*$^{CreERT2/+}$ mice (set as 1). *N* = 3 biological replicates/genotype (VCS cardiomyocytes pooled from 30 mice per biological replicate); multiple testing correction using Benjamini and Hochberg procedure. Significance was assessed by Welch *t*-test (*FDR <0.05; **FDR <0.005; ***FDR <0.001) and confirmed by one-tailed Wilcoxon test (p value in parentheses) when normal distribution was rejected. Western blotting, n=2 biological replicates/ genotype (VCS cardiomyocytes pooled from 50 mice per each biological replicate). Abbreviations: AVB, atrioventricular bundle (also known as bundle of His); AVN, atrioventricular node; dCKO, double-conditional knockout; FDR, false discovery rate; LBB, left bundle branches; RBB, right bundle branches; VCS, ventricular conduction system.

The online version of this article includes the following source data for figure 4:

**Source data 1.** File containing original western blots corresponding to *Figure 4D*.

**Source data 2.** Zipped folder containing original western blots files corresponding to *Figure 4D*.

Our previous research (*Arnolds et al., 2012*; *Moskowitz et al., 2007*; *Burnicka-Turek et al., 2020*; *Moskowitz et al., 2004*) and published literature *van Duijvenboden et al., 2014*; *Bakker et al., 2012*; *Bruneau et al., 2001*; *Frank et al., 2011*; *Mohan et al., 2020*; *van Eif et al., 2018* have suggested that the balance between *Tbx3* and T*bx5* expression determines the regional specialization of the mature central CCS (*Figure 7*). *Tbx3* expression dominates in nodal myocardium, imparting nodal physiological characteristics. *Tbx5* expression dominates in fast VCS myocardium, where a T-box-dependent fast conduction system network drives physiologically dominant fast conduction

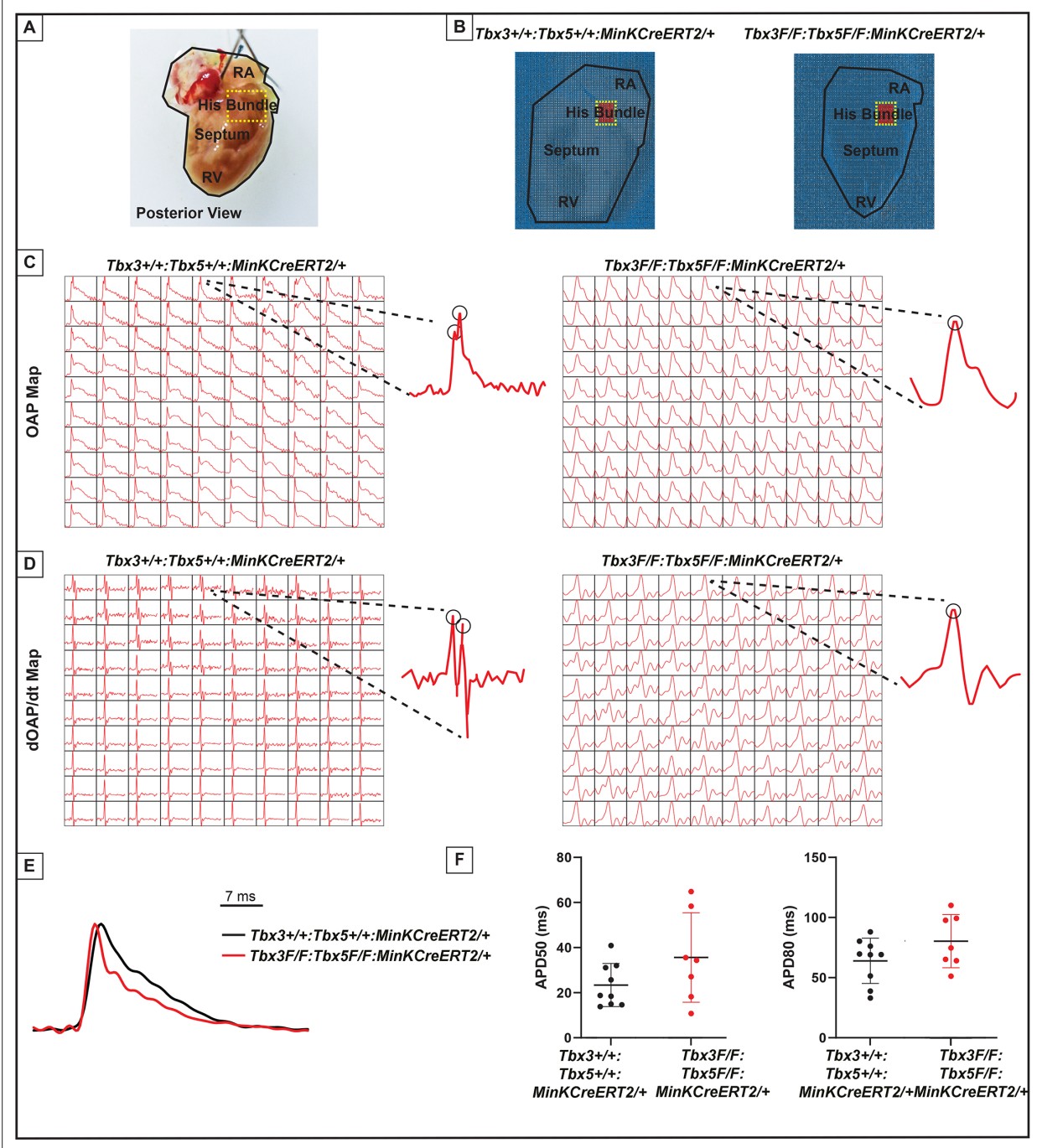

**Figure 5.** Loss of VCS optical action potential (OAP) morphology in *Tbx3:Tbx5* double-conditional knockout mice. (**A**) Schematic of the posterior view of a mouse heart with right ventricle (RV) free wall removed, highlighting the RV, septum, His bundle, and right atria (RA). (**B**) Representative 100 × 100 OAP map of OAP recorded during sinus rhythm from *Tbx3:Tbx5* double-conditional knockout mice and control littermates with the free wall removed. The region of the His bundle is highlighted in red. (**C**) Representative 10 × 10 OAP map from the region of the His bundle. (**D**) Representative 10 × 10 map of the first derivative of the OAP from the region of the His bundle. (**E**) Representative ventricular OAP from whole heart intact preparation from *Tbx3:Tbx5* double-conditional knockout mice (red) and control littermates (black). (**F**) Quantification of APD50 and APD80 at a basic cycle length of 125 ms. For (C-F), n=10 or 8 animals/ genotype.

The online version of this article includes the following figure supplement(s) for figure 5:

**Figure supplement 1.** *Tbx3:Tbx5* double-conditional knockout mice exhibit QRS prolongation.

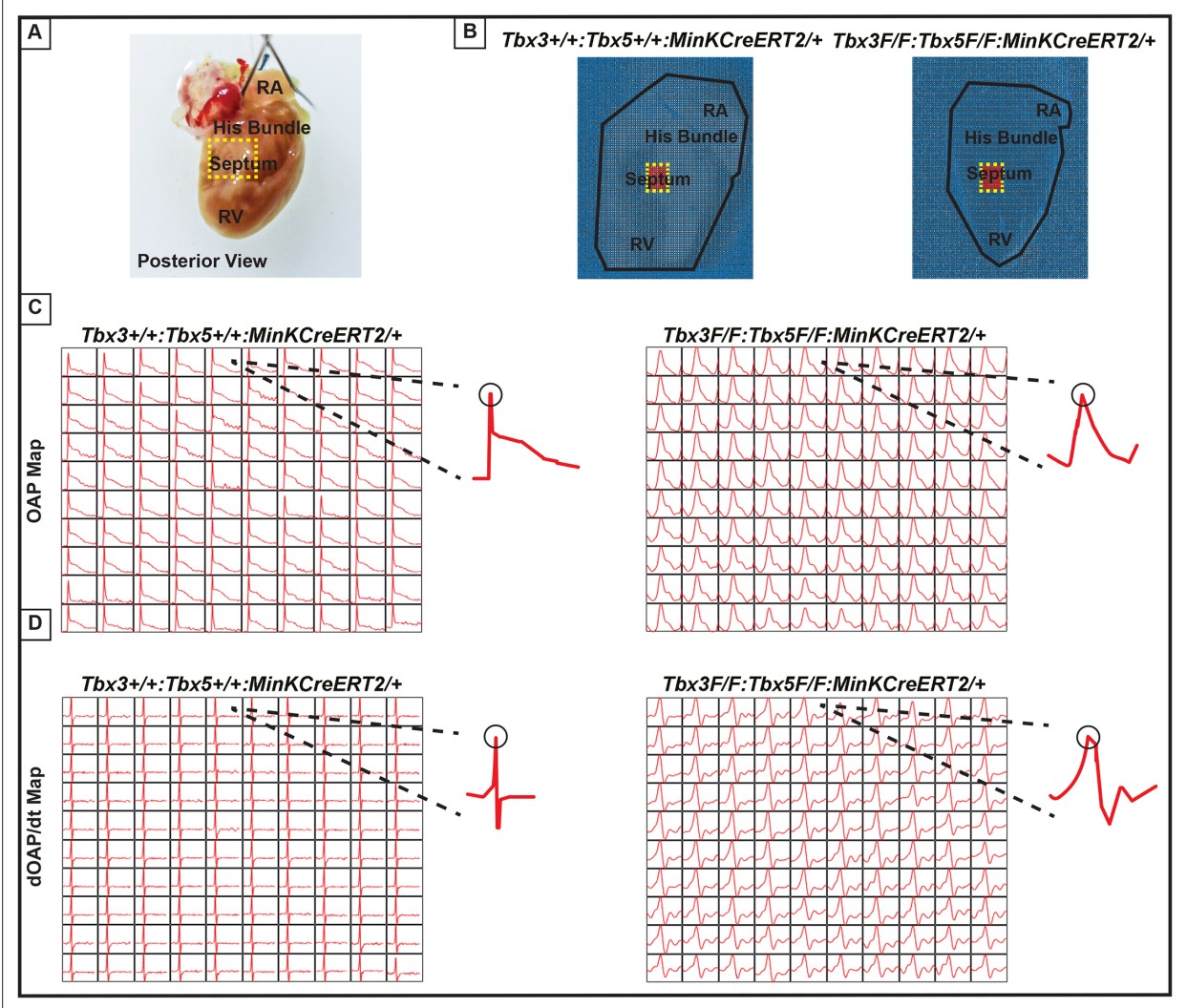

**Figure 6.** Ventricular optical action potentials (OAPs) distal from His bundle have only one OAP upstroke. (**A**) Schematic of the posterior view of mouse heart with right ventricle (RV) free wall removed. (**B**) Representative 100 × 100 pixel OAP map recorded during sinus rhythm from *Tbx3:Tbx5* double-conditional knockout mice and control littermates with RV free wall removed. The region of the working ventricular myocardium distal from the His bundle is highlighted in red. (**C**) Representative 10 × 10 pixel OAP map from the region distal to the His bundle. (**D**) Representative 10 × 10 dOAP/dt map from the region distal to the His bundle. For (C and D), n=10 or 8 animals/ genotype.

physiology, overriding nodal physiology (***Burnicka-Turek et al., 2020***; ***Figure 7***). This model accurately predicts the outcome of targeted manipulations, such as adult VCS-specific removal of TBX5 or overexpression of TBX3, which convert the fast VCS into a slow nodal-like system (***Burnicka-Turek et al., 2020***; ***Figure 7***).

The adult CCS organization model posits that *Tbx3* and *Tbx5* coordinately establish CCS characteristics and suggests their compound necessity to maintain specialized VCS identity (***Figure 7***), a hypothesis that remained untested. This model predicted that VCS-specific genetic ablation of both the TBX3 and TBX5 transcription factors would transform fast-conducting adult VCS into cells resembling working myocardium, eliminating specialized CCS identity (***Figure 7***). Testing this model necessitated the generation of a *Tbx3:Tbx5* double-conditional knockout allele due to their proximal chromosomal location in cis on mouse Chr5. VCS-specific double-knockout mice showed a notable deceleration in VCS conduction, manifested by prolonged PR and QRS intervals, along with increased susceptibility to VT. A similar functional phenotype was observed in single deletion of *Tbx5* from the adult VCS, leading to the transformation of the fast VCS into a nodal-like phenotype (***Arnolds et al., 2012***; ***Burnicka-Turek et al., 2020***). However, we predicted that the nodal-like characteristics of the

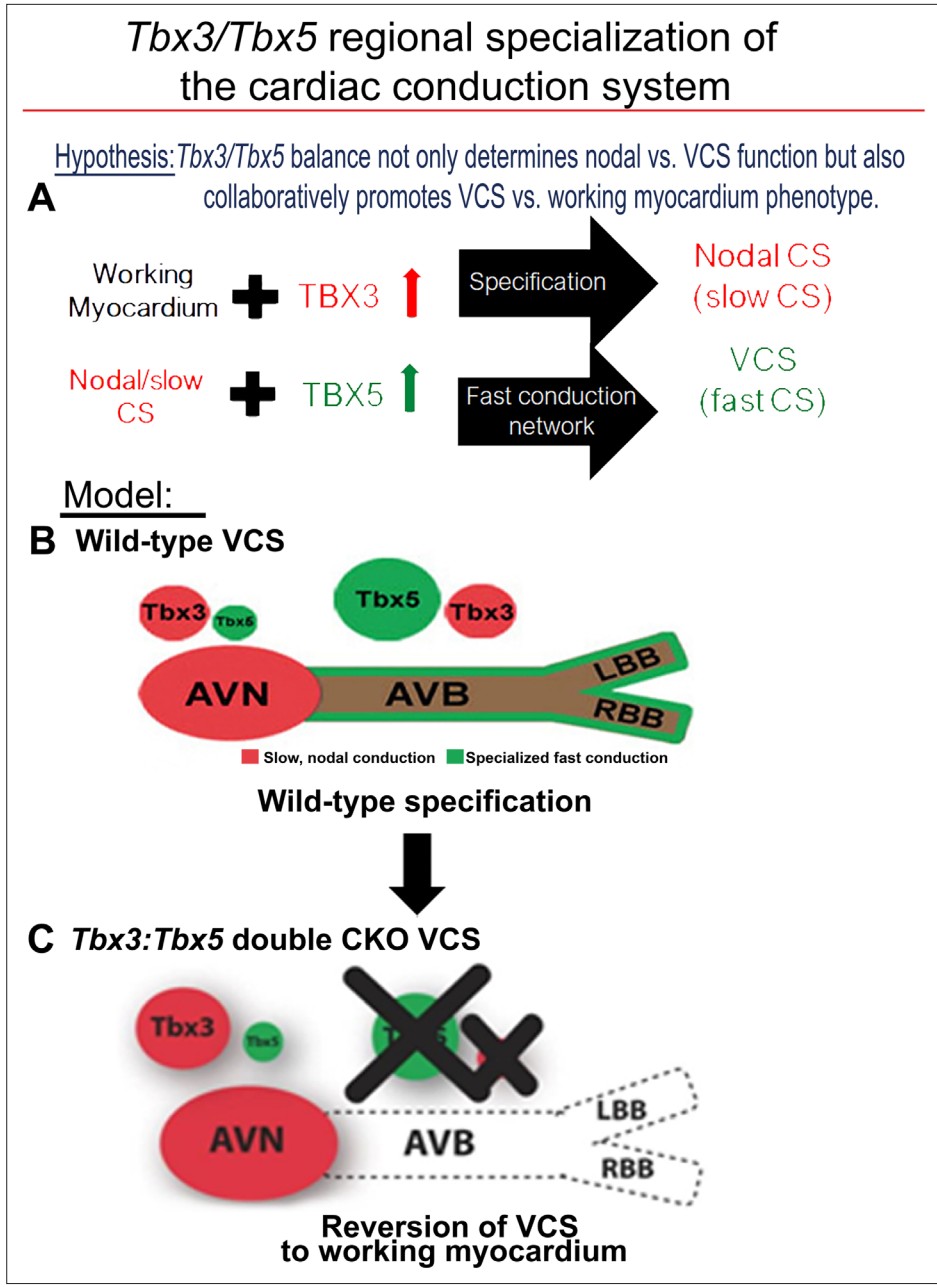

**Figure 7.** *Tbx3* and *Tbx5* play distinct roles in the adult VCS while collectively promoting CCS regional identity – a model elucidating our hypothesis for *Tbx3/Tbx5* dose-dependent CCS regional specialization. (**A**) The *Tbx3/Tbx5* balance not only governs nodal versus ventricular conduction system (VCS) function but also collaboratively promotes the VCS versus working myocardium (WM) phenotype. Specifically, a high level of *Tbx3* is linked to the specialization to the nodal conduction system, while an elevated *Tbx5* level in nodal cells activates local expression of the *Tbx5*-dependent fast conduction network, resulting in the generation of VCS. (**B**) CCS regional specialization is driven by local expression of *Tbx5*-dependent fast conduction network in the VCS, which overlaps underlying Pan-CCS expression of nodal, slow conduction network. (**C**) VCS-specific simultaneous genetic removal of both the *Tbx3* and *Tbx5* transcription factors transforms the fast-conducting, adult VCS into cells resembling working myocardium, thereby shifting them from conduction to non-conduction myocytes. Therefore, within the adult CCS, the *Tbx3* and *Tbx5* expression levels are crucial not only for normal fast versus slow conduction system identity but also for maintaining the conduction versus contraction specialization of the VCS. AVB, atrioventricular bundle; AVN, atrioventricular node; CCS, cardiac conduction system; CKO, conditional knockout; LBB, left bundle branch; RBB, right bundle branch; VCS, ventricular conduction system.

*Tbx5*-ablated VCS were due to retained expression of *Tbx3*. In fact, the autonomous beating and impulse initiation observed in the *Tbx5*-mutant VCS (*Arnolds et al., 2012*; *Burnicka-Turek et al., 2020*) were absent from the double *Tbx3:Tbx5* mutant VCS, further suggesting the transformation from a nodal-like to an inert myocardial functionally and a shift from conduction toward simple working myocardium.

Molecular studies support a transformation from conduction to working myocardium in the *Tbx3:Tbx5* double VCS knockout. Previous studies investigating the roles of *Tbx3* and *Tbx5* in maintaining adult VCS identity demonstrated that *Tbx3* deletion resulted in the silencing of Pan-CCS gene expression in the atrioventricular conduction system (*van Duijvenboden et al., 2014*; *Bakker et al., 2012*; *Bruneau et al., 2001*; *Frank et al., 2011*; *Mohan et al., 2020*). Alternately, specific deletion of *Tbx5* from the adult VCS led to the decreased expression of VCS-specific markers while Pan-CCS markers remained unchanged, indicating a transformation toward a nodal-like transcriptional phenotype in the absence of *Tbx5* (*Arnolds et al., 2012*; *Burnicka-Turek et al., 2020*). Consistently, a similar transformation was induced by *Tbx3* overexpression in the adult VCS. These findings underscored the significance of the *Tbx3:Tbx5* ratio in preserving the molecular and functional characteristics of the fast versus slow conduction system. In *Tbx3:Tbx5* double VCS knockout, we observed a reduction in the expression of both fast VCS markers and Pan-CCS markers transcribed throughout the entire CCS. As expected, not all VCS markers were ablated in VCS-specific *Tbx3:Tbx5* mutants. A significant portion of VCS fast conduction markers are also transcribed in ventricular working myocardium, albeit at lower levels than in the VCS, and are crucial for normal working myocardial function, for example, *Ryr2*, *Scn5a*, and *Kcnj2* (*van Eif et al., 2020*; *Schram et al., 2002*; *Remme et al., 2009*). Consistent with a shift from fast VCS to working myocardium, their expression levels are present, but at significantly lower levels in the double-knockout mutants than in normal VCS. Furthermore, the expression of markers specific for working myocardium, which are normally excluded in the VCS, emerged in the VCS of *Tbx3:Tbx5* double mutant. These observations are consistent with a transcriptional shift from VCS conduction cardiomyocytes to working myocardium-like characteristics in the absence of both *Tbx3* and *Tbx5*.

Our study highlights the critical role of *Tbx3* and *Tbx5* in maintaining the specialized electrical properties of the VCS. Comparative analysis of the phenotypes observed in single *Tbx3* or *Tbx5* knockouts and the *Tbx3:Tbx5* double-conditional knockout supports a coordinated function for these transcription factors in preserving VCS functionality. In a normal mammalian heart, electrical or optical recordings from the proximal part of the VCS typically display two distinct electrical excitations, reflected as two spikes in electrograms or two upstrokes in OAPs. These signals correspond to the sequential activation of the VCS, followed by ventricular excitation, a phenomenon well documented in both basic and clinical studies across species, including mice and humans. Previous whole-heart EP and optical mapping studies demonstrated that the single conditional knockout of *Tbx5* in the adult VCS caused significant ventricular conduction slowing (*Arnolds et al., 2012*), accompanied by a phenotypic shift of VCS cells toward a pacemaker-like cell, evidenced by ectopic beats originating in the ventricles, retrograde activation, and inappropriate automaticity (*Burnicka-Turek et al., 2020*). Whole-cell patch clamp recordings of *Tbx5*-deficient VCS cells further supported these findings, revealing action potential characteristics resembling pacemaker cells, including a slower upstroke (phase 0), prolonged plateau (phase 2), delayed repolarization (phase 3), and enhanced phase 4 depolarization (*Burnicka-Turek et al., 2020*). In contrast, reduced expression of *Tbx3* in the *Tbx3* haploinsufficiency model resulted in AV bundle hypoplasia, PR interval shortening, and prolonged QRS duration (*Mohan et al., 2020*). Optical mapping in these mice showed delayed apical activation with multiple small breakthroughs separated by regions of delayed activation. This fragmented activation pattern suggests that the reduced *Tbx3* expression impaired the homogeneity and efficiency of ventricular electrical impulse propagation due to AV bundle and bundle branch hypoplasia (*Mohan et al., 2020*). In the *Tbx3:Tbx5* double-conditional knockout mice, we observed a complete loss of fast VCS conduction, similar to the phenotype of the *Tbx5* single conditional knockouts. However, unlike in the *Tbx5* single CKO, the VCS in *Tbx3:Tbx5* double mutants did not acquire pacemaker-like activity. Instead, optical mapping revealed a single upstroke in the His bundle region of *Tbx3:Tbx5* double-conditional knockout mice, indicating the absence of rapid AV bundle electrical impulse propagation. Functionally, this transformation resulted in significantly slowed ventricular activation, without the retrograde activation observed in the *Tbx5* single conditional knockouts or the rapid electrical

impulse propagation typically associated with an intact VCS. The homogenous morphological and molecular transformation of His bundle and VCS cells in the *Tbx3:Tbx5* double-conditional knockout mice rendered the electrical properties of the VCS-located cells functionally indistinguishable from those of ventricular working myocardium. Our data, including gene expression analysis of the double mutant VCS, are most consistent with the transition of double mutant VCS cells toward a working myocardium phenotype.

These findings emphasize the interdependent roles of *Tbx3* and *Tbx5* in maintaining the specialized electrical properties and identity of the VCS, distinguishing it from non-specialized ventricular working myocardium (*Figure 7*). While *Tbx5* alone is critical for suppressing pacemaker-like characteristics and preserving fast conduction properties, *Tbx3* plays a pivotal role in ensuring the structural and functional integrity of the AV and BB conduction pathways. The simultaneous deletion of both *Tbx3* and *Tbx5* disrupts these regulatory mechanisms, leading to a loss of VCS identity and a functional shift toward a working myocardium phenotype. Our study combined with prior literature (*Arnolds et al., 2012*; *van Duijvenboden et al., 2014*; *Burnicka-Turek et al., 2020*; *Bakker et al., 2012*; *Frank et al., 2011*; *Mohan et al., 2020*; *van Eif et al., 2018*) indicates that the concurrent presence of both *Tbx3* and *Tbx5* is necessary for maintaining VCS identity in the adult heart, ensuring efficient electrical conduction and preventing the transition of VCS cells into a working myocardium-like state (*Figure 7*).

## Methods
### Experimental animals
All animal experiments were performed under the University of Chicago Institutional Animal Care and Use Committee (IACUC) approved protocol (ACUP no. 71737) and in compliance with the USA Public Health Service Policy on Humane Care and Use of Laboratory Animals. *MinK$^{CreERT2}$* [Tg(RP23-276I20-*MinKCreERT2*)] and *Tbx3$^{fl/fl}$* mice have been reported previously (*Arnolds and Moskowitz, 2011b*; *Frank et al., 2011*).

*Tbx3:Tbx5* double-floxed mouse line was generated by University of Utah Core Research Facility using CRISPR/Cas9. Guide RNA (sgRNA) constructs were designed with software tools (ZiFiT Targeter *Sander et al., 2010* and crispr.genome-engineering.org) predicting unique target sites throughout the mouse genome (*Figure 1—figure supplement 3–5*). The sgRNA constructs were transcribed in vitro using MEGAshortscript T7 (Invitrogen AM1354) and mMessage Machine T7 transcription kit (Invitrogen AM1344) according to the manufacturer's instructions. The strategy to generate mouse founders involved a single-step microinjection into one-cell *Tbx3* floxed zygotes (on a mixed background) with 10 ng/μl of each sgRNA (*Tbx5*-I2-S22 and *Tbx5*-I3-S31, *Figure 1—figure supplements 3–5*), 30 ng/μl of Cas9 protein, and 10 ng/μl of a long ssDNA donor. The donor contained two lox2272 sites in cis, spanning partial intron 2, entire exon 3, and partial intron 3 of the mouse *Tbx5* gene, along with 100/150 bp 5′/3′ homology arms (*Figure 1—figure supplement 4*). Founders were validated by PCR, restriction enzyme digestion, and Sanger sequencing (*Figure 1—figure supplement 1B–D*). Founders were backcrossed with wild-type CD1 IGS mice (Charles River Lab, USA) to confirm germline transmission of the CRISPR/Cas9-generated compound *Tbx3:Tbx5* double-floxed allele and obtain the F1 generation. F1 mice were then interbred to establish a stable *Tbx3:Tbx5* double-floxed mouse line. Downstream experiments were performed on F4–F6 mice.

To simultaneously conditionally delete the *Tbx3* and *Tbx5* genes specifically from the adult VCS, we crossed our *Tbx3:Tbx5* double-floxed mouse line (*Tbx3$^{fl/fl}$;Tbx5$^{fl/fl}$*) with a VCS-specific tamoxifen (TM)-inducible *Cre* transgenic mouse line (*MinK$^{CreERT2}$* [Tg(RP23-276I20-*MinKCreERT2*); *Arnolds and Moskowitz, 2011b*; *Figure 1A*]). All mice were maintained on a mixed genetic background. Tamoxifen (MP Biomedical) was administered at a dose of 0.167 mg/g body weight for 5 consecutive days by oral gavage at 6 weeks of age and then mice were evaluated at 9 weeks of age, as previously described (*Arnolds et al., 2012*; *Arnolds and Moskowitz, 2011b*; *Burnicka-Turek et al., 2020*). Age-, gender-, and genetic strain-matched controls were used in all experiments to account for any variations in datasets across experiments. Mice were bred and housed in specific pathogen-free conditions in a 12-hr light/12-hr dark cycle and allowed ad libitum access to standard mouse chow and water. Mice requiring medical attention were provided with appropriate veterinary care by a licensed veterinarian and were excluded from the experiments described. No other exclusion criteria were applied.

All experiments and subsequent analysis were conducted in a blinded fashion, with animals randomly assigned to experimental groups. Following genotyping, mice were randomly allocated to the studies based on their genotypes. Subsequently, their identities were anonymized using a numerical code to ensure that all experiments and analyses were performed in a blinded manner. Both male and female animals have been used in our studies in the ratio of 41%/59%, based on availability of relatively rare compound genotypes, respectively.

## Validation of the newly generated Tbx5 floxed allele

The *Tbx3:Tbx5* double-floxed mouse line (*Tbx3^{fl/fl}:Tbx5^{fl/fl}*) was engineered by generating a novel *Tbx5* floxed allele in the background of a previously published *Tbx3* floxed allele (**Frank et al., 2011**). We utilized single *Tbx3* (**Frank et al., 2011**) and the novel *Tbx5* floxed mouse lines (*Tbx3^{fl/fl}* and *Tbx5^{fl/fl}*, respectively) to serve as controls in our studies (**Figure 1A**, **Figure 1—figure supplements 1 and 2**). Although the newly engineered *Tbx5* floxed allele was designed to mirror a previously published allele (**Bruneau et al., 2001**), the location of the flox sites was altered slightly, requiring validation.

To verify the ability of the newly generated *Tbx5* floxed allele to recombine into the *Tbx5* KO (null) allele, we conducted three rounds of breeding and utilized a PCR assay to distinguish the recombined *Tbx5* KO (null) allele (*Tbx5⁻*) from the unrecombined floxed allele (*Tbx5^{fl}*) (**Figure 1—figure supplement 2A, B**). First, we bred newly generated homozygous *Tbx5* floxed males (*Tbx5^{fl/fl}*) with homozygous *Mef2C^{Cre/Cre}* females, which express *Cre* recombinase at the zygote stage (**Hoffmann et al., 2014**; **Xie et al., 2012**; **Verzi et al., 2005**), to produce germline *Tbx5^{+/−};Mef2C^{Cre/+}* double heterozygous mice. PCR analysis showed that 100% of the generated offspring were *Tbx5^{+/−};Mef2C^{Cre/+}* double heterozygous mice (**Figure 1—figure supplement 2B**). This result confirmed complete recombination of the new *Tbx5* floxed allele into the *Tbx5* null allele in the presence of the *Mef2C^{Cre}* allele. Next, we backcrossed *Tbx5^{+/−};Mef2C^{Cre/−}* double heterozygote mice with wild-type CD1 IGS mice (Charles River Lab, USA) to obtain a germline *Tbx5^{+/−}* heterozygous mouse line (**Figure 1—figure supplement 2B**). Subsequently, we bred *Tbx5^{+/−}* mice to generate germline *Tbx5* null mice (*Tbx5^{−/−}*). Among 230 offspring (generated in 50 litters), we observed 113 wild-type pups (*Tbx5^{+/+}*) and 116 *Tbx5* heterozygous pups (Tbx5^{+/−}), but no null newborn pups (*Tbx5^{−/−}*) were obtained (**Figure 1—figure supplement 2B**). These results align with previous breeding outcomes indicating complete embryonic lethality of *Tbx5* null mice (**Bruneau et al., 2001**).

To further confirm the ability of the newly generated *Tbx5* floxed allele to produce germline *Tbx5* null embryos (*Tbx5^{−/−}*), we bred *Tbx5* heterozygous mice and collected embryos at E9–E10. Genotyping analysis of 89 collected embryos from 10 timed pregnancies revealed the presence of 24 wild-type (*Tbx5^{+/+}*), 42 heterozygous (*Tbx5^{+/−}*), and 22 null (*Tbx5^{−/−}*) embryos (**Figure 1—figure supplement 2C**). Supported by Chi-square analysis, these results collectively verified that the newly generated *Tbx5* floxed allele could recombine to form the null allele, mimicking the embryonic lethality observed with the previously generated allele (**Bruneau et al., 2001**).

To confirm that the newly engineered *Tbx5* conditional allele can generate VCS-specific *Tbx5*-deficient mice with the same phenotype as VCS-specific *Tbx5* knockout mice obtained from the previously generated allele (**Arnolds et al., 2012**; **Burnicka-Turek et al., 2020**; **Bruneau et al., 2001**), we crossed the newly generated *Tbx5* floxed mouse line (*Tbx5^{fl/fl}*) with a VCS-specific tamoxifen-inducible Cre transgenic mouse line (*MinK^{CreERT2}* [Tg(RP23-276I20-*MinKCreERT2*)]) (**Arnolds and Moskowitz, 2011b**; **Figure 1A**). The same tamoxifen regimen was administered as in the previous study (**Arnolds et al., 2012**; **Burnicka-Turek et al., 2020**). We confirmed VCS-specific genetic deletion of *Tbx5* via immunofluorescence (IF) (**Figure 1B**) and qPCR (**Figure 1C**). The resulting VCS-specific *Tbx5*-deficient mice exhibited slowed conduction, indicated by increased PR and QRS intervals (**Figure 2A, and E vs. B**), as well as episodes of VT resulting in sudden death (**Figure 2G, H**), consistent with the previously reported phenotype (**Arnolds et al., 2012**; **Burnicka-Turek et al., 2020**). Additionally, we confirmed the absence of histological and structural abnormalities in these mice, aligning with previous findings (**Figure 3A, F vs. B, and K vs. G**, respectively) (**Arnolds et al., 2012**; **Burnicka-Turek et al., 2020**).

## Echocardiography studies

Transthoracic echocardiography in mice was conducted under inhaled isoflurane anesthesia administered through a nose cone. Prior to imaging, chest hairs were removed using a topical depilatory agent. Limb leads were affixed for electrocardiogram gating, and animals were imaged in the left

lateral decubitus position with a VisualSonics Vevo 770 machine using a 30-MHz high-frequency transducer. To ensure stability, body temperature was carefully maintained using a heated imaging platform and warming lamps. Two-dimensional images were meticulously recorded in parasternal long- and short-axis projections, accompanied by guided M-mode recordings at the midventricular level in both views. LV cavity size and percent FS were measured in at least three beats from each projection and averaged. M-mode measurements were employed to ascertain LV chamber dimensions and percent LV FS, calculated as ([LVIDd – LVIDs]/LVIDd), where LVIDd and LVIDs represent LV internal diameter in diastole and systole, respectively.

## Surface ECG

Nine weeks old, tamoxifen-treated control and mutant mice were anesthetized with a mixture of 2–3% isoflurane in 100% oxygen. Anesthetized mice were secured in a supine position on a regulated heat pad while lead I and lead II ECGs were recorded using platinum subdermal needle electrodes in a three-limb configuration. Core temperature was continuously monitored using a rectal probe and maintained at 36–37°C throughout the procedure. ECG data were collected and analyzed using Ponemah Physiology Platform (DSI) software and an ACQ-7700 acquisition interface unit (Gould Instruments, Valley View, OH, USA). Key parameters derived from the ECG measurements included: heart rate (HR), PR interval (from the beginning of the P wave to the beginning of the QRS complex), and QRS complex duration.

## Telemetry ECG analysis

Nine weeks old, tamoxifen-treated control and mutant mice were anesthetized with 2–3% isoflurane in 100% oxygen, and wireless telemetry transmitters (ETA-F10; DSI) were surgically implanted in the back with leads tunneled to the right upper and left lower thorax, as previously described (*Arnolds et al., 2012*; *Wheeler et al., 2004*). Following a 24-hr recovery period after surgical instrumentation, heart rate and PR and QRS intervals were calculated using Ponemah Physiology Platform (DSI) from 48-hr recordings.

## Catheter-based intracardiac EP

Detailed protocols for invasive EP studies have been previously described (*Nadadur et al., 2016*; *Liu et al., 2008*). Briefly, 9 weeks old, tamoxifen-treated control and mutant mice were anesthetized using 2–3% isoflurane in 100% oxygen. Then, a 1.1-Fr octapolar electrode catheter (EPR-800; Millar Instruments) was advanced via a right jugular venous cut-down to record right atrial, His bundle, and RV potentials, as well as to perform programmed electrical stimulation. Signals were identified through alignment with simultaneous surface ECG using subcutaneous needle electrodes in a lead II configuration. 'Near-Field' and 'Far-Field' signals were identified based on ECG alignment, signal deflection upstroke speed, and total signal duration. Standard tachycardia induction protocols included an eight-beat drive train with beats 80–120 ms apart (S1), followed by five beats (S2) at 50 ms apart (penta-extrastimulus, PES). Two attempts at this PES protocol were carried out. Mice also underwent single extrastimulus testing (SES) with eight beats 80–120 ms apart (S1) followed by a single S2 at 50 ms. This SES protocol was carried out five separate times per mouse. If these two protocols in the right atrium and RV (separately) failed to initiate their respective tachycardias, the study was deemed negative. S1 drive train intervals varied slightly due to the presence of the AV Wenckebach block at faster pacing rates in some mice; the S1 interval was lengthened to prevent this during the drive train.

Additional atrial and ventricular pacing protocols were carried out to obtain atrial, atrio-ventricular, and ventricular effective refractory periods (AERP, AVERP, and VERP) as well as sinus node recovery time, as described previously (*Nadadur et al., 2016*; *Liu et al., 2008*; *Patel et al., 2003*; *Gehrmann and Berul, 2000*). Effective refractory periods were measured using eight-beat S1 drive trains of 100 ms followed by SES.

## ECG and optical mapping

### ECG acquisition and analysis

ECGs were recorded in conscious *Tbx3:Tbx5* double-conditional knockout mice and control littermates using the ecgTUNNEL device (emka Technologies) 2 weeks after tamoxifen treatment. Mice were positioned in the tunnel and ECGs were recorded for 5 min at a sampling rate of 1 kHz using lead

I. ECGs were analyzed using a custom MATLAB program to measure P and QRS durations, as well as PR, QT, and RR intervals (*George et al., 2020*; *Warhol et al., 2021*; *Figure 5—figure supplement 1*).

## Langendorff perfusion

*Tbx3:Tbx5* double-conditional knockout mice ($n$ = 8) and control littermates ($n$ = 10) were deeply anesthetized with isoflurane. The heart was quickly excised following cervical dislocation and thoracotomy. The aorta was cannulated, and the heart was retrogradely perfused with warmed (37°C) and oxygenated (95% $O_2$ and 5% $CO_2$) modified Tyrode's solution (in mM, NaCl 130, $NaHCO_3$ 24, $NaH_2PO_4$ 1.2, $MgCl_2$ 1, glucose 5.6, KCl 4, and $CaCl_2$ 1.8) at a pH of 7.4. The heart was placed in a constant-flow (1.0–2.0 ml/min) Langendorff perfusion system, laying horizontally in a tissue bath.

## Ventricular optical mapping

The Langendorff perfused heart was electromechanically uncoupled by 15 µM of blebbistatin (Cayman Chemicals 13186) perfusion. A voltage-sensitive fluorescent dye, 80 µM di-4-ANEPPs (Thermo Fisher Scientific D1199), was administered through the dye injection port. The heart was illuminated using a 520 ± 5 nm (Prizmatix, UHP-Mic-LED-520) wavelength light source to excite di-4-ANEPPs. Emitted photons were captured using complementary metal–oxide semiconductor cameras (MiCAM, SciMedia).

The stimulation threshold, where the heart would capture the stimuli and action potential 1:1, was determined using a point source platinum electrode placed on the anterior surface of the heart on the anterior epicardial surface of the ventricles. Pacing was applied at 1.5× the threshold amplitude to maintain 1:1 capture over the duration of the experiment. Optical recordings were analyzed using Rhythm 1.2 to analyze transmembrane potential (*Cathey et al., 2019*). Action potential duration at 50% and 80% repolarization was calculated for the ventricles of the intact whole heart preparation.

## Right septal preparation optical mapping experiments

Following intact whole heart ex vivo optical mapping, the heart was removed from the tissue bath for the dissection of the RV free wall to expose the RV septal surface (*Tamaddon et al., 2000*). The heart was quickly returned to the tissue bath and the RV septal surface was focused into the field of view. Sinus rhythm optical recordings were acquired.

A custom MATLAB program was written to plot individual pixels in a 100 × 100 pixel image stack. A Butterworth filter of order 5 was applied to provide temporal filtering (*Laughner et al., 2012*). The location of the His bundle was identified as the region at the base of the interventricular septum (a red 10 × 10 pixel area in *Figure 5B*). Signals from the working ventricular myocardium were recorded from the 10x10 pixel region plotted in the working ventricular myocardium distal to the His bundle (*Figure 6B*).

## Isolation of adult VCS cardiomyocytes and cell sorting

Adult mouse EYFP-positive VCS cardiomyocytes were isolated using the method described by *Mitra and Morad, 1985*. Briefly, 9 weeks old tamoxifen-treated control and *Tbx3:Tbx5* double-conditional mutant mice were heparinized (100 units IP) and anesthetized with pentobarbital (50 mg/kg of body weight). Hearts were excised and mounted on a Langendorff apparatus, then perfused with $Ca^{2+}$-free Tyrode's solution with collagenase B and D (Roche Chemical Co) plus protease (Fraction IV, Sigma Chemical Co). When the hearts appeared pale and flaccid, they were removed from the Langendorff apparatus and a tip of ventricular septum below the AV annulus was microdissected out (*Arnolds et al., 2012*; *Park and Fishman, 2011*; *Silverman et al., 2006*) and kept in $Ca^{2+}$-free Tyrode's solution with 1 mg/ml of bovine serum albumin (Fraction XIV, Sigma Chemical Co). The intraventricular sections were then minced into small pieces ~1 × 1 × 1 mm and then gently triturated with a Pasteur pipette to dissociate individual VCS myocytes. Propidium iodide (Thermo Fisher Scientific) was added immediately before FACS to facilitate live/dead discrimination. Cells were sorted on a FacsAria flow cytometer (BD Biosciences) located at the University of Chicago Flow Cytometry Core using Influx software. Samples from wild-type age-matched hearts were used for gating. Samples were gated to exclude debris and cell clumps. Fluorescent cells were collected into ice-cold RNase-free PBS and processed for RNA extraction and protein isolation.

## RNA isolation and qRT-PCR

Total RNA was isolated from EYFP-positive VCS cardiomyocytes sorted from 9-week-old control and VCS-specific *Tbx3:Tbx5*-deficient mice, which had received tamoxifen at 6 weeks of age. RNA was also extracted from atrial and ventricular tissues dissected from the same mice. All RNA extractions were performed using RNeasy Mini Kit (QIAGEN), followed by DNase treatment according to the manufacturer's instructions.

Reverse transcription reaction was carried out using the SuperScript III First-Strand Synthesis SuperMix for quantitative RT-PCR (Invitrogen) as per the manufacturer's recommendations. qRT-PCR was performed using the POWER SYBR Green PCR master mix from Applied Biosystems and run on an Applied Biosystems AB7500 machine in 96-well plates. The relative gene expression level was calculated by the ΔΔCt method (*Livak and Schmittgen, 2001*) using glyceraldehyde-3-phosphate dehydrogenase (*Gapdh*) gene expression level as internal control. The data presented are the average of three independent experiments.

## Protein isolation and western blotting

Total protein was isolated from EYFP-positive VCS cardiomyocytes sorted from 9-week-old control and VCS-specific *Tbx3:Tbx5*-deficient mice that had been administered tamoxifen at 6 weeks of age. Protein was also obtained from atrial and ventricular tissues dissected from the same animals. The tissues were snap-frozen in liquid nitrogen, pulverized, and homogenized in RIPA buffer (50 mM Tris-HCl pH 8, 150 mM NaCl, 1% Triton-X, 0.5% sodium deoxycholate, 0.1% SDS, 5 mM EDTA) with 1 Roche EDTA-Free complete protease inhibitor tablet per 50 ml of buffer. Samples were tumbled for 1 hr at 4°C and then centrifuged for 10 min at 13,200 × *g*. Protein concentration was determined using the BCA assay (Pierce) with BSA as a standard.

For Western blot analysis, 25 μg of protein was diluted in Laemmli buffer, heated at 70°C for 10 min, and subjected to SDS–PAGE on 4–20% TGX gels (Bio-Rad). Proteins were then transferred to nitrocellulose membranes, blocked with 5% milk in TBS-T, and incubated overnight at 4°C with primary antibodies diluted in 2.5% milk in TBS-T. The primary antibodies used were: goat polyclonal anti-TBX3 (Santa Cruz Biotechnology, sc-31656, 1:250), rabbit polyclonal anti-HCN4 (Millipore, AB5808, 1:500), rabbit polyclonal anti-CAV1.3/CACNA1D (Alomone, ACC-005, 1:200), rabbit polyclonal anti-Cx45/GJC1 (Thermo Fisher Scientific, PA5-77357, 1:250), sheep polyclonal anti-TBX5 (R&D, AF5918, 1:200), rabbit polyclonal anti-CX40/GJA5 (Zymed/Invitrogen, 36-4900, 1:500), rabbit polyclonal anti-NAV1.5/SCN5A (Alomone, ASC-005, 1:200), mouse monoclonal anti-KCNK3/TASK1 (Abcam, ab186352, 1:1000), rabbit polyclonal anti-CX43/GJA1 (Cell Signaling Technology, 3512, 1:1000), and mouse monoclonal anti-GAPDH (Abcam, ab8245, 1:1000). After rinsing in TBS-T, membranes were incubated for 1 hr at room temperature (RT) with secondary antibodies diluted in 2.5% milk in TBS-T, rinsed again, and visualized using enhanced chemiluminescence reagents (Pierce ECL/ECL Plus, Thermo Fisher Scientific) and Kodak X-OMAT film. Results were normalized to GAPDH loading control and then quantified using ImageJ software (*Rasband, 1997*; *Wheeler et al., 2004*). Secondary antibodies used were donkey anti-goat IgG AlexaFluor-594 (Invitrogen, A-11058, 1:250 dilution) and donkey anti-goat IgG AlexaFluor-488 (Invitrogen, A-11055, 1:250 dilution) for experiments involving co-staining for goat primary antibodies. Secondary antibodies were as follows: rabbit anti-goat-HRP (Jackson ImmunoResearch, 305-035-003, 1:10,000), goat anti-rabbit-HRP (Jackson ImmunoResearch, 111-035-144, 1:3000), donkey anti-sheep-HRP (Abcam, ab6900, 1:5000), and sheep anti-mouse-HRP (Amersham GE, NA931, 1:2500).

## Histology

Hearts from 9-week-old control and mutant mice were carefully dissected, flushed with ice-cold PBS, and then fixed for 48 hr at 4°C in a 4% paraformaldehyde solution (Sigma-Aldrich). Following fixation, the hearts were processed for paraffin-embedded sections and subjected to analysis through H&E staining, following the manufacturer's protocol provided by Sigma-Aldrich.

## Immunofluorescence

Hearts from 9-week-old control and mutant mice were dissected out and placed in ice-cold PBS, followed by freezing in OCT (Fisher) within the gas phase of liquid nitrogen. Cryosections of 7 μm thickness were mounted onto Superfrost Plus glass slides (Fisher Scientific), air-dried, and fixed for

10 min in ice-cold 4% paraformaldehyde (Sigma-Aldrich). Subsequently, sections were permeabilized in 1% Triton-X100 (Sigma-Aldrich) in PBS for 10 min, then blocked in 10% normal goat serum (Invitrogen) in PBS-T (PBS + 0.1% Tween-20) for 30 min at RT. Sections were then incubated overnight at 4°C in primary antibody diluted in blocking buffer, rinsed in PBS, then subsequently incubated for 60 min at RT in secondary antibody diluted in blocking buffer. Slides were mounted in VectaShield + DAPI (Vector Laboratories) or counterstained with DAPI and mounted in ProLog Gold (Invitrogen) prior to visualizing fluorescence. Primary antibodies were as follows: goat polyclonal anti-TBX3 (Santa Cruz, sc-31656, 1:250 dilution) and goat polyclonal anti-TBX5 (Santa Cruz, sc-17866, 1:250 dilution). Secondary antibodies used were donkey anti-goat IgG AlexaFluor-594 (Invitrogen, A-11058, 1:250 dilution) and donkey anti-goat IgG AlexaFluor-488 (Invitrogen, A-11055, 1:250 dilution) for experiments involving co-staining for goat primary antibodies. A secondary antibody-only control was employed in each case to ensure the specificity of immunostaining.

## Statistics

The numbers of independent experiments are specified in the relevant figure legends. Quantitative data are presented as mean ± SD. Statistical analysis was performed using R version 3.6.2 or the GraphPad Prism statistical package version 10.2.1 (GraphPad Software, Boston, MA, USA). For all datasets, except qRT-PCR data, normality was assessed using the Shapiro–Wilk normality test. If the data followed a normal distribution, the two-sample Welch *t*-test was performed for comparison. If normal distribution was not present, the non-parametric Mann–Whitney *U* test or Kruskal–Wallis *H* test was used, as indicated in the text. Statistical significance was assumed if p reached a value ≤0.05.

For qRT-PCR data, statistical analysis was performed using R version 4.2.0. Normality within each experimental group was assessed using the Shapiro–Wilk test. Between-group comparisons were conducted using Welch *t*-test, and multiple comparisons were corrected using the Benjamini and Hochberg method to control the false discovery rate (FDR) (*Benjamini and Hochberg, 1995*). If a significant difference was detected between two groups (*t*-test FDR <0.05) but normality was rejected in any of the compared groups (Shapiro–Wilk p < 0.05), a non-parametric Wilcoxon rank-sum test was used for verification. A significant group-mean difference was confirmed at one-tailed Wilcoxon p ≤ 0.05 (*Source data 1*).

## Acknowledgements

This work was supported by National Institutes of Health (R01 HL163523 and R01 HL148719 to IP Moskowitz), Leducq Foundation Grant Bioelectronics for Neuroradiology – Diagnosis & Therapeutics (to IR Efimov), and American Heart Association (13POST17290028 to O Burnicka-Turek).

## Additional information

### Funding

| Funder | Grant reference number | Author |
| --- | --- | --- |
| National Institutes of Health | R01 HL163523 | Ivan P Moskowitz |
| National Institutes of Health | R01 HL148719 | Ivan P Moskowitz |
| Leducq Foundation Grant Bioelectronics for Neuroradiology – Diagnosis & Therapeutics | | Igor R Efimov |
| American Heart Association | 13POST17290028 | Ozanna Burnicka-Turek |

The funders had no role in study design, data collection, and interpretation, or the decision to submit the work for publication.

## Author contributions
Ozanna Burnicka-Turek, Conceptualization, Data curation, Formal analysis, Supervision, Validation, Investigation, Visualization, Methodology, Writing – original draft, Project administration, Writing – review and editing; Katy A Trampel, Conceptualization, Data curation, Formal analysis, Validation, Investigation, Visualization, Methodology, Writing – original draft, Writing – review and editing; Brigitte Laforest, Data curation, Formal analysis, Validation, Investigation, Visualization; Michael T Broman, Resources, Data curation, Formal analysis, Supervision, Validation, Investigation, Visualization, Methodology; Xinan H Yang, Data curation, Formal analysis, Visualization, Writing – review and editing; Zoheb Khan, Margaret Gadek, Kaitlyn M Shen, Data curation, Validation, Investigation; Eric Rytkin, Binjie Li, Investigation, Methodology; Ella Schaffer, Data curation, Investigation, Visualization; Igor R Efimov, Conceptualization, Resources, Formal analysis, Supervision, Funding acquisition, Methodology, Writing – original draft, Project administration, Writing – review and editing; Ivan P Moskowitz, Conceptualization, Resources, Supervision, Funding acquisition, Writing – original draft, Project administration, Writing – review and editing

## Author ORCIDs
Ozanna Burnicka-Turek ⬤ https://orcid.org/0000-0002-3770-4314
Katy A Trampel ⬤ https://orcid.org/0000-0001-5735-1037
Brigitte Laforest ⬤ https://orcid.org/0000-0001-6919-8922
Igor R Efimov ⬤ https://orcid.org/0000-0002-1483-5039
Ivan P Moskowitz ⬤ https://orcid.org/0000-0003-0014-4963

## Ethics
All animal experiments were performed under the University of Chicago Institutional Animal Care and Use Committee (IACUC) approved protocol (ACUP no. 71737) and in compliance with the USA Public Health Service Policy on Humane Care and Use of Laboratory Animals.

Reviewer #2 (Public review): https://doi.org/10.7554/eLife.102027.3.sa1
Reviewer #3 (Public review): https://doi.org/10.7554/eLife.102027.3.sa2
Author response https://doi.org/10.7554/eLife.102027.3.sa3

---

# Additional files

## Supplementary files
Source data 1. Expanded statistical analysis of the qRT-PCR data presented in *Figures 1C, 3L and 4A–C*. Statistical analyses were performed in R (v4.2.0). Normality for each experimental group was assessed using the Shapiro-Wilk test. Group comparisons were conducted using Welch's t-test, with multiple testing correction by the Benjamini & Hochberg method to control the false discovery rate (FDR) (74). If FDR < 0.05 but normality was rejected (Shapiro-Wilk P < 0.05), significance was confirmed using a one-tailed Wilcoxon rank-sum test (P ≤ 0.05).

MDAR checklist

## Data availability
All data generated and analyzed during this study are included in the main manuscript and its supplementary materials. The source data for statistical analyses are provided in Source Data 1. Uncropped gels and blots have been supplied for Figure 1-figure supplements 1 and 2, as well as for Figure 4. No additional data are required to interpret the findings.

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

# Appendix 1

## Appendix 1—key resources table

| Reagent type (species) or resource | Designation | Source or reference | Identifiers | Additional information |
|---|---|---|---|---|
| Antibody | Goat polyclonal anti-TBX3 | Santa Cruz Biotechnology | sc-31656 | IF (1:250) WB (1:250) |
| Antibody | Goat polyclonal anti-TBX5 | Santa Cruz Biotechnology | sc-17866 | IF (1:250) |
| Antibody | Donkey anti-goat IgG AlexaFluor-594 | Invitrogen | A-11058 | IF (1:250) |
| Antibody | Donkey anti-goat IgG AlexaFluor-488 | Invitrogen | A-11055 | IF (1:250) |
| Antibody | Rabbit polyclonal anti-HCN4 | Millipore | AB5808 | WB (1:500) |
| Antibody | Rabbit polyclonal anti-CAV1.3/CACNA1D | Alomone | ACC-005 | WB (1:200) |
| Antibody | Rabbit polyclonal anti-Cx45/GJC1 | Thermo Fisher | PA5-77357 | WB (1:250) |
| Antibody | Sheep polyclonal anti-TBX5 | R&D | AF5918 | WB (1:200) |
| Antibody | Rabbit polyclonal anti-CX40/GJA5 | Zymed/ Invitrogen | 36–4900 | WB (1:500) |
| Antibody | Rabbit polyclonal anti-NAV1.5/SCN5A | Alomone | ASC-005 | WB (1:200) |
| Antibody | Mouse monoclonal anti-KCNK3/TASK1 | Abcam | ab186352 | WB (1:1000) |
| Antibody | Rabbit polyclonal anti-CX43/GJA1 | Cell Signalling Technology | 3512 | WB (1:1000) |
| Antibody | Mouse monoclonal anti-GAPDH | Abcam | ab8245 | WB (1:1000) |
| Antibody | Rabbit anti-goat-HRP | Jackson Immuno Research | 305-035-003 | WB (1:10 000) |
| Antibody | Goat anti-rabbit-HRP | Jackson Immuno Research | 111-035-144 | WB (1:3000) |
| Antibody | Donkey anti-sheep-HRP | Abcam | ab6900 | WB (1:5000) |
| Antibody | Sheep anti-mouse-HRP | Amersham GE | NA931 | WB (1:2500) |
| Chemical compound, drug | BSA | Sigma-Aldrich | A9576 | |
| Chemical compound, drug | cOmplete, Mini, EDTA-free Protease Inhibitor Cocktail | Roche | 11836170001 | |
| Chemical compound, drug | DAPI | Invitrogen | 62248 | |
| Chemical compound, drug | Isoflurane | Sigma-Aldrich | 26675-46-7 | |
| Chemical compound, drug | POWER SYBR Green PCR Master Mix | Applied Biosystems | 43-676-59 | |
| Chemical compound, drug | ProLong Gold Antifade Mountant | Invitrogen | P10144 | |
| Chemical compound, drug | Propidium Iodide | Thermo Fisher Scientific | P3566 | |
| Chemical compound, drug | Tamoxifen | MP Biomedical | | Dosage details look at Materials & Methods section |
| Commercial assay or kit | MEGAshortscript T7 transcription kit | Invitrogen | AM1354 | |
| Commercial assay or kit | mMessage Machine T7 transcription kit | Invitrogen | AM1344 | |

*Appendix 1 Continued on next page*

*Appendix 1 Continued*

| Reagent type (species) or resource | Designation | Source or reference | Identifiers | Additional information |
|---|---|---|---|---|
| Commercial assay or kit | Pierce BCA Protein Assay Kit | Thermo Fisher Scientific | 23225 | |
| Commercial assay or kit | Pierce ECL | Thermo Fisher Scientific | 32106 | |
| Commercial assay or kit | Pierce ECL Plus | Thermo Fisher Scientific | 32132 | |
| Commercial assay or kit | RNeasy Mini Kit | Qiagen | 74104 | |
| Commercial assay or kit | SuperScript III First-Strand Synthesis SuperMix | Invitrogen | 18080400 | |
| Genetic reagent (*M. musculus*) | Tbx3$^{fl/fl}$;Tbx5$^{fl/fl}$ Mixed 129/SvJ:C57BL/6 J:CD-1 background | This paper | | Generated using CRISPR–Cas9 technology (look at Experimental animals section) |
| Genetic reagent (*M. musculus*) | Tbx3$^{fl/fl}$;Tbx5$^{fl/fl}$; R26R$^{eYFP/+}$; MinK$^{CreERT2/+}$ Mixed 129/SvJ:C57BL/6 J:CD-1 background | This paper | | Generated using CRISPR–Cas9 technology (look at Experimental animals section) |
| Genetic reagent (*M. musculus*) | Kcne1$^{CreERT2}$ [Tg(RP23-276I20-MinKCreERT2)] – referred to as MinK$^{CreERT2}$ in the manuscript for consistency with field-standard nomenclature and to avoid confusion. Mixed 129/SvJ:C57BL/6 J:CD-1 background | PMID: 21504046 | | Dr. Ivan P. Moskowitz (University of Chicago) |
| Genetic reagent (*M. musculus*) | Tbx3$^{fl/fl}$ Mixed 129/SvJ:C57BL/6 J:CD-1 background | PMID: 22203979 | | Dr. Anne M Moon (University of Utah) |
| Genetic reagent (*M. musculus*) | CD-1 | Charles Rivers Laboratories | | https://www.criver.com/products-services/find-model/cd-1r-igs-mouse?region=3611 |
| Genetic reagent (*M. musculus*) | C57Bl/6 J | Jackson Laboratory | | https://www.jax.org/strain/000664 |
| Other | Long ssDNA donor | This paper | | Sequence details look at *Figure 1—figure supplement 4* |
| Other | SDS-PAGE on 4–20% TGX Gels | Bio-Rad | 4561096 | 4–20% Mini-PROTEAN TGX Precast Protein Gels |
| Other | Tbx5-I2-S22 | This paper | sgRNA | Sequence details look at *Figure 1—figure supplement 3* |
| Other | Tbx5-I2-S31 | This paper | sgRNA | Sequence details look at *Figure 1—figure supplement 3* |
| Other | VectaShield +DAPI | Vector Laboratories | | https://vectorlabs.com/vectashield-mounting-medium-with-dapi.html |
| Sequence-based reagent | Tbx5 locus: 5'arm-F | This paper | PCR primers | Sequence details look at *Figure 1—figure supplement 5* |
| Sequence-based reagent | Tbx5 locus: 5'arm-R | This paper | PCR primers | Sequence details look at *Figure 1—figure supplement 5* |
| Sequence-based reagent | Tbx5 locus: 3'arm-F | This paper | PCR primers | Sequence details look at *Figure 1—figure supplement 5* |
| Sequence-based reagent | Tbx5 locus: 3'arm-R | This paper | PCR primers | Sequence details look at *Figure 1—figure supplement 5* |
| Sequence-based reagent | Tbx3 locus: WT-F | PMID: 22203979 | PCR primers | Dr. Anne M Moon (University of Utah) |

*Appendix 1 Continued on next page*

*Appendix 1 Continued*

| Reagent type (species) or resource | Designation | Source or reference | Identifiers | Additional information |
|---|---|---|---|---|
| Sequence-based reagent | Tbx3 locus: WT-R | PMID: 22203979 | PCR primers | Dr. Anne M Moon (University of Utah) |
| Sequence-based reagent | Tbx3 locus: Flox-F | PMID: 22203979 | PCR primers | Dr. Anne M Moon (University of Utah) |
| Sequence-based reagent | Tbx3 locus: Flox-R | PMID: 22203979 | PCR primers | Dr. Anne M Moon (University of Utah) |
| Sequence-based reagent | qPCR_Tbx3-F | PMID: 32290757 | qRT-PCR primers | 5'-AGATCCGGTT ATCCCTGG GAC-3' |
| Sequence-based reagent | qPCR_Tbx3-R | PMID: 32290757 | qRT-PCR primers | 5'-CAGCAGCCCC CACTAACTG-3' |
| Sequence-based reagent | qPCR_Tbx5-F | PMID: 32290757 | qRT-PCR primers | 5'-GGCATGGAAG GAATCAAG GT-3' |
| Sequence-based reagent | qPCR_Tbx5-R | PMID: 32290757 | qRT-PCR primers | 5'-CTAGGAAACATT CTCCTCCC TGC-3' |
| Sequence-based reagent | qPCR_Gja1-F | PMID: 22086960 (Primer Bank ID:166091435 c3) | qRT-PCR primers | 5'-ACAGCGGTTG AGTCAGCT TG-3' |
| Sequence-based reagent | qPCR_Gja1-R | PMID: 22086960 (Primer Bank ID:166091435 c3) | qRT-PCR primers | 5'-GAGAGATGGG GAAGGACT TGT-3' |
| Sequence-based reagent | qPCR_Gja5-F | PMID: 32290757 | qRT-PCR primers | 5'-AGCTCCAGTC ACCCATCT TG-3' |
| Sequence-based reagent | qPCR_Gja5-R | PMID: 32290757 | qRT-PCR primers | 5'-CAGTTGAACA GCAGCCAG AG-3' |
| Sequence-based reagent | qPCR_Gjc1-F | PMID: 32290757 | qRT-PCR primers | 5'-AGATCCACAA CCATTCGA CATTT-3' |
| Sequence-based reagent | qPCR_Gjc1-R | PMID: 32290757 | qRT-PCR primers | 5'-TCCCAGGTAC ATCACAGA GGG-3' |
| Sequence-based reagent | qPCR_Gjd3-F | PMID: 22086960 (Primer Bank ID: 30519904 a1) | qRT-PCR primers | 5'-TCATGCTGAT CTTCCGCA TCC-3' |
| Sequence-based reagent | qPCR_Gjd3-R | PMID: 22086960 (Primer Bank ID: 30519904 a1) | qRT-PCR primers | 5'-GAAGCGGTAG TGGGACACC-3' |
| Sequence-based reagent | qPCR_Scn5a-F | PMID: 32290757 | qRT-PCR primers | 5'-CGCTCCTCCA GGTAGATG TC-3' |
| Sequence-based reagent | qPCR_Scn5a-R | PMID: 32290757 | qRT-PCR primers | 5'-CTACCGCATA GTGGAGCA CA-3' |
| Sequence-based reagent | qPCR_Ryr2-F | PMID: 32290757 | qRT-PCR primers | 5'-CAAATCCTTC TGCTGCCA AG-3' |
| Sequence-based reagent | qPCR_Ryr2-R | PMID: 32290757 | qRT-PCR primers | 5'-CGAGGATGAG ATCCAGTT CC-3' |
| Sequence-based reagent | qPCR_Kcnk3-F | PMID: 32290757 | qRT-PCR primers | 5'-CTCCTTCTAC TTCGCCAT CA-3' |
| Sequence-based reagent | qPCR_Kcnk3-R | PMID: 32290757 | qRT-PCR primers | 5'-GAAGGTGTTG ATGCGTTCA-3' |
| Sequence-based reagent | qPCR_Kcnj2-F | PMID: 32290757 | qRT-PCR primers | 5'-CGACTGCCAT GACAACTC AA-3' |
| Sequence-based reagent | qPCR_Kcnj2-R | PMID: 32290757 | qRT-PCR primers | 5'-CATATCTCCG ATTCTCGCCT-3' |
| Sequence-based reagent | qPCR_Kcnj3-F | PMID: 32290757 | qRT-PCR primers | 5'-GCTGGCAACT ACACTCCC TG-3' |

*Appendix 1 Continued on next page*

*Appendix 1 Continued*

| Reagent type (species) or resource | Designation | Source or reference | Identifiers | Additional information |
|---|---|---|---|---|
| Sequence-based reagent | qPCR_Kcnj3-R | PMID: 32290757 | qRT-PCR primers | 5'-AACATGCAGC CGATGAGG AA-3' |
| Sequence-based reagent | qPCR_Kcnj4-F | PMID: 32290757 | qRT-PCR primers | 5'-CACGTAAACG GCTTTTTG GGG-3' |
| Sequence-based reagent | qPCR_Kcnj4-R | PMID: 32290757 | qRT-PCR primers | 5'-CCGTCTCGAA CGGAGATG AC-3' |
| Sequence-based reagent | qPCR_Kcnj12-F | PMID: 32290757 | qRT-PCR primers | 5'-ACCCCTACAG CATCGTAT CAT-3' |
| Sequence-based reagent | qPCR_Kcnj12-R | PMID: 32290757 | qRT-PCR primers | 5'-GTTGCACTGA CCGTTCTT CTT-3' |
| Sequence-based reagent | qPCR_Hcn1-F | PMID: 22086960 (Primer Bank ID: 6754168 a1) | qRT-PCR primers | 5'-CAAATTCTCC CTCCGCAT GTT-3' |
| Sequence-based reagent | qPCR_Hcn1-R | PMID: 22086960 (Primer Bank ID: 6754168 a1) | qRT-PCR primers | 5'-TGAAGAACGT GATTCCAA CTGG-3' |
| Sequence-based reagent | qPCR_Hcn4-F | PMID: 32290757 | qRT-PCR primers | 5'-GGCGGACACC GCTATCAAA-3' |
| Sequence-based reagent | qPCR_Hcn4-R | PMID: 32290757 | qRT-PCR primers | 5'-TGCCGAACAT CCTTAGGG AGA-3' |
| Sequence-based reagent | qPCR_Cacna1d-F | PMID: 22086960 (Primer Bank ID: 27413155 a1) | qRT-PCR primers | 5'-GACTGATGCC CGATATAA AGGC-3' |
| Sequence-based reagent | qPCR_Cacna1d-R | PMID: 22086960 (Primer Bank ID: 27413155 a1) | qRT-PCR primers | 5'-CCTTCACCAGA AATAGGGA GTCT-3' |
| Sequence-based reagent | qPCR_Cacna1g-F | PMID: 32290757 | qRT-PCR primers | 5'-TGTCTCCGCA CGGTCTGT AA-3' |
| Sequence-based reagent | qPCR_Cacna1g-R | PMID: 32290757 | qRT-PCR primers | 5'-AGATACCCAA AGCGACCA TCTT-3' |
| Sequence-based reagent | qPCR_Cacna1h-F | PMID: 32290757 | qRT-PCR primers | 5'-GAACGTGGTT CTTTACAA CGGC-3' |
| Sequence-based reagent | qPCR_Cacna1h-R | PMID: 32290757 | qRT-PCR primers | 5'-GCACATAGTT CCCAAAGG TCA-3' |
| Sequence-based reagent | qPCR_Smpx-F | PMID: 22086960 (Primer Bank ID: 14149752 a1) | qRT-PCR primers | 5'-ATGTCGAAGC AGCCAATT TCC-3' |
| Sequence-based reagent | qPCR_Smpx-R | PMID: 22086960 (Primer Bank ID: 14149752 a1) | qRT-PCR primers | 5'-TCAGACAAGT TGACAACA GGTC-3' |
| Sequence-based reagent | qPCR_Col1a1-F | PMID: 22086960 (Primer Bank ID: 34328108 a1) | qRT-PCR primers | 5'-GCTCCTCTTA GGGGCCACT-3' |
| Sequence-based reagent | qPCR_Col1a1-R | PMID: 22086960 (Primer Bank ID: 34328108 a1) | qRT-PCR primers | 5'-CCACGTCTCA CCATTGGGG-3' |
| Sequence-based reagent | qPCR_Postn-F | PMID: 22086960 (Primer Bank ID: 7657429 a1) | qRT-PCR primers | 5'-CCTGCCCTTA TATGCTCT GCT-3' |
| Sequence-based reagent | qPCR_Postn-R | PMID: 22086960 (Primer Bank ID: 7657429 a1) | qRT-PCR primers | 5'-AAACATGGTCA ATAGGCAT CACT-3' |

