## [Editor Report · eLife Assessment]

The work presented is **important** for our understanding of the development of the cardiac conduction system and its regulation by T-box transcription factors. The conclusions are supported by **convincing** data. Overall this is an excellent study that advances our understanding of cardiac biology and has implications beyond the immediate field of study.

---

## [Referee Report · Reviewer #2 (Public review)]

Summary:

The goal of this work is to define the functions of T-box transcription factors Tbx3 and Tbx5 in the adult mouse ventricular cardiac conduction system (VCS) using a novel conditional mouse allele in which both genes are targeted in cis. A series of studies over the past 2 decades by this group and others have shown that Tbx3 is a transcriptional repressor that patterns the conduction system by repressing genes associated with working myocardium, while Tbx5 is a potent transcriptional activator of "fast" conduction system genes in the VCS. In a previous work, the authors of the present study further demonstrated that Tbx3 and Tbx5 exhibit an epistatic relationship whereby the relief of Tbx3-mediated repression through VCS conditional haploinsufficiency allows better toleration of Tbx5 VCS haploinsufficiency. Conversely, excess Tbx3-mediated repression through overexpression results in disruption of the fast-conduction gene network despite normal levels of Tbx5. Based on these data the authors proposed a model in which repressive functions of Tbx3 drive adoption of conduction system fate, followed by segregation into a fast-conducting VCS and slow-conduction AVN through modulation of the Tbx5/Tbx3 ratio in these respective tissue compartments.

The question motivating the present work is: If Tbx5/Tbx3 ratio is important for slow versus fast VCS identity, what happens when both genes are completely deleted from the VCS? Is conduction system identity completely lost without both factors and if so, does the VCS network transform into a working myocardium-like state? To address this question, the authors have generated a novel mouse line in which both Tbx5 and Tbx3 are floxed on the same allele, allowing complete conditional deletion of both factors using the VCS-specific MinK-CreERT2 line, convincingly validated in previous work. The goal is to use these double conditional knockout mice to further explore the model of Tbx3/Tbx5 co-dependent gene networks and VCS patterning. First the authors demonstrate that the double conditional knockout allele results in the expected loss of Tbx3 and Tbx5 specifically in the VCS when crossed with Mink-CreERT2 and induced with tamoxifen. The double conditional knockout also results in premature mortality. Detailed electrophysiological phenotyping demonstrated prolonged PR and QRS intervals, inducible ventricular tachycardia, and evidence of abnormal impulse propagation along the septal aspect of the right ventricle. In addition, the mutants exhibit downregulation of VCS genes responsible for both fast conduction AND slow conduction phenotypes with upregulation of 2 working myocardial genes including connexin-43. The authors conclude that loss of both Tbx3 and Tbx5 results in "reversion" or "transformation" of the VCS network to a working myocardial phenotype, which they further claim is a prediction of their model and establishes that Tbx3 and Tbx5 "coordinate" transcriptional control of VCS identity.

Overall Appraisal:

As noted above, the present study does not further explore the Tbx5/Tbx3 ratio concept since both genes are completely knocked out in the VCS. Instead, the main claims are that absence of both factors results in a transcriptional shift of conduction tissue towards a working myocardial phenotype, and that this shift indicates that Tbx5 and Tbx3 "coordinate" to control VCS identity and function. However, only limited data are presented to support the claim of transcriptional reprogramming since the knockout cells are not directly compared to working myocardial cells at the transcriptional level and only a small number of key genes are assessed (versus genome-wide assessment). In addition, the optical mapping dataset has alternative interpretations that are not excluded or thoroughly discussed.

In sum, while this study adds an elegantly constructed genetic model to the field, the data presented mostly fit within the existing paradigm of established functions of Tbx3 and Tbx5. The authors present some evidence to support the claim that VCS cells adopt a working myocardial phenotype in the absence of Tbx3 and Tbx5, but some key experiments that could more definitively test this model were not performed, reducing the degree to which the data support the conclusions.

Strengths:

(1) Successful generation of a novel Tbx3-Tbx5 double conditional mouse model

(2) Successful VCS-specific deletion of Tbx3 and Tbx5 using a VCS-specific inducible Cre driver line

(3) Well-powered and convincing assessments of mortality and physiological phenotypes

(4) Isolation of genetically modified VCS cells using flow.

Weaknesses:

(1) In general, the data is consistent with a long-standing and well-supported model in which Tbx3 represses working myocardial genes and Tbx5 activates expression of VCS genes, which seem like distinct roles in VCS patterning.

(2) More direct quantitative comparison of Tbx5 Adult VCS KO with Tbx5/Tbx3 Adult VCS double KO would be helpful to ascertain whether deletion of Tbx3 on top of Tbx5 deletion changes the underlying phenotype in some discernable way beyond mRNA expression of a few genes. Superficially, the phenotypes look quite similar at the EKG and arrhythmia inducibility level and no optical mapping data from single Tbx5 KO is presented for comparison to the double KO. I understand that single Tbx5 VCS KO mutants have been evaluated in previous publications but I think in order to evaluate the claims presented here, it would be important to do a direct comparison using the same assays and conditions.

(3) The authors claim that double knockout VCS cells transform to working myocardial fate, but there is no comparison of gene expression levels between actual working myocardial cells and the Tbx3/Tbx5 DKO VCS cells so it's hard to know if the data reflect an actual cell state change or a more non-specific phenomenon with global dysregulation of gene expression or perhaps dedifferentiation. I understand that the upregulation of Gja1 and Smpx is intended to address this, but it's only two genes and it seems relevant to understand their degree of expression relative to actual working myocardium. In addition, the gene panel is somewhat limited and does not include other key transcriptional regulators in the VCS such as Irx3 and Nkx2-5. RNA-seq in these populations would provide a clearer comparison among the groups.

(4) From the optical mapping data, it is difficult to distinguish between the presence of (1) a focal proximal right bundle branch block due to dysregulation of gene expression in the VCS but overall preservation of the right bundle and its distal ramifications; from (2) actual loss of the VCS with reversion of VCS cells to a working myocardial fate. Related to this, the authors claim that this experiment allows for direct visualization of His bundle activation, but can the authors confirm or provide evidence that the tissue penetration of their imaging modality allows for imaging of a deep structure like the AV bundle as opposed to the right bundle branch which is more superficial? Does the timing of the separation of the sharp deflection from the subsequent local activation suggest visualization of more distal components of the VCS rather than the AV bundle itself? Additional clarification would be helpful.

impact:

The present study contributes a novel and elegantly constructed mouse model to the field. The data presented generally corroborate existing models of transcriptional regulation in the VCS. Acknowledging that the present work is strong start, some additional studies not included in the present manuscript will be needed for this new mouse model to decisively advance the field of VCS transcriptional biology.

---

## [Referee Report · Reviewer #3 (Public review)]

Summary:

In the study presented by Burnicka-Turek et al., the authors generated for the first time a mouse model to cause the combined conditional deletion of Tbx3 and Tbx5 genes. This has been impossible to achieve to date due to the proximity of these genes in chromosome 5, preventing the generation of loss of function strategies to delete simultaneously both genes. It is known that both Tbx3 and Tbx5 are required for the development of the cardiac conduction system by transcription factor-specific but also overlapping roles as seen in the common and diverse cardiac defects found in patients with mutations for these genes. After validating the deletion efficiency and specificity of the line, the authors characterised the cardiac phenotype associated to cardiac conduction system (CCS)-specific combined deletion of Tbx5 and Tbx3 in the adult by inducing the activation of the CCS-specific tamoxifen inducible Cre recombination (MinK-creERT) at 6 weeks after birth. Their analysis of 8-9 weeks old animals did not identify any major morphological cardiac defects. However, the authors found conduction defects including prolonged PR and QTR intervals and ventricular tachycardia causing the death of the double mutants, which do not survive more than 3 months after tamoxifen induction. Molecular and optical mapping analysis of the ventricular conduction system (VCS) of these mutants concluded that, in the absence of Tbx5 and Tbx3 function, the cells forming the ventricular conduction system (VCS) become working myocardium and lose the specific contractile features characterising VCS cells. Altogether, the study identified the critical combined role of Tbx3 and Tbx5 in the maintenance of the VCS in adulthood.

Strengths:

The study generated a new animal model to study the combined deletion of Tbx5 and Tbx3 in the cardiac conduction system. This unique model has provided the authors with the perfect tool to answer their biological questions. The study includes top-class methodologies to assess the functional defects present in the different mutants analysed, and gathered very robust functional data on the conduction defects present in these mutants. They also applied optical action potential (OAP) methods to demonstrate the loss of conduction action potential and the acquisition of working myocardium action potentials in the affected cells because of Tbx5/Tbx3 loss of function. The study used simpler molecular and morphological analysis to demonstrate that there are no major morphological defects in these mutant and that indeed, the conduction defects found are due to the acquisition of working myocardium features by the VCS cells. Altogether, this study identified the critical role of these transcription factors in the maintenance of the VCS in the adult heart.

Weaknesses:

In the opinion of this reviewer, the weakness in the study lays in the morphological and molecular characterization. The morphological analysis simply described the absence of general cardiac defects in the adult heart, however, whether the CCS tissues are present or not was not investigated. Linage tracing analysis using the reporter lines included in the crosses described in the study, will determine if there are changes in CCS tissue composition in the different mutants studied. Similarly, combining this reporter analysis with the molecular markers found to be dysregulated by qPCR and western blot will demonstrate that indeed the cells that were specified as VCS in the adult heart become working myocardium in the absence of Tbx3 and Tbx5 function.

Comments on revisions:

I would like to thank the authors for their revised manuscript and for their corrections based on the suggestions from the 3 reviewers. Although I would have preferred to see some of the additional experiments suggested by any of the reviewers to improve the robustness and depth of the study integrated in the revised version of the manuscript, I acknowledge that the authors may prefer to develop them as follow-up studies. So, looking forward to seeing the follow-up study unravelling the detailed molecular regulation controlled by Tbx3/Tbx5 during the formation and maintenance of the ventricular cardiac conduction system.

---

## [Author Response]

The following is the authors’ response to the original reviews

**Reviewer #1 (Public review):**
Summary:In a heroic effort, Ozanna Burnicka-Turek et al. have made and investigated conduction system-specific *Tbx3-Tbx5* deficient mice and investigated their cardiac phenotype. Perhaps according to expectations, given the body of literature on the function of the two T-box transcription factors in the heart/conduction system, the cardiomyocytes of the ventricular conduction system seemed to convert to "ordinary" ventricular working myocytes. As a consequence, loss of VCS-specific conduction system propagation was observed in the compound KO mice, associated with PR and QRS prolongation and elevated susceptibility to ventricular tachycardia.Strengths:Great genetic model. Phenotypic consequences at the organ and organismal levels are well investigated. The requirement of both *Tbx3* and *Tbx5* for maintaining VCS cell state has been demonstrated.

We thank Reviewer #1 for acknowledging the effort involved in generating and characterizing the *Tbx3/Tbx5* double conditional knockout mouse model and for highlighting the significance of this work in elucidating the role of these transcription factors in maintaining the functional and transcriptional identity of the ventricular conduction system.

Weaknesses:The actual cell state of the *Tbx3/Tbx5* deficient conducting cells was not investigated in detail, and therefore, these cells could well only partially convert to working cardiomyocytes, and may, in reality, acquire a unique state.

We agree with Reviewer #1 that the *Tbx3/Tbx5* double mutant ventricular conduction myocardial cells may only partially convert to working cardiomyocytes or may acquire a unique state. The transcriptional state of the double mutant VCS cells was investigated by bulk profiling of key genes associated with specific conduction and non-conduction cardiac regions, including fast conduction, slow conduction, or working myocardium. Neither the bulk transcriptional approaches nor the optical mapping approaches we employed capture single-cell data; in both cases, the data represents aggregated signals from multiple cells (1, 2). Single cell approaches for transcriptional profiling and cellular electrophysiology would clarify this concern and are appropriate for future studies.

(1) O’Shea C, Nashitha Kabri S, Holmes AP, Lei M, Fabritz L, Rajpoot K, Pavlovic D (2020) Cardiac optical mapping – State-of-the-art and future challenges. The International Journal of Biochemistry & Cell Biology 126:105804. doi: 10.1016/j.biocel.2020.105804. (2) Efimov IR, Nikolski VP, and Salama G (2004) Optical Imaging of the Heart. Circulation Research 95:21-33. doi: 10.1161/01.RES.0000130529.18016.35.

**Reviewer #2 (Public review):**
Summary:The goal of this work is to define the functions of T-box transcription factors *Tbx3* and *Tbx5* in the adult mouse ventricular cardiac conduction system (VCS) using a novel conditional mouse allele in which both genes are targeted in cis. A series of studies over the past 2 decades by this group and others have shown that *Tbx3* is a transcriptional repressor that patterns the conduction system by repressing genes associated with working myocardium, while *Tbx5* is a potent transcriptional activator of "fast" conduction system genes in the VCS. In a previous work, the authors of the present study further demonstrated that *Tbx3* and *Tbx5* exhibit an epistatic relationship whereby the relief of *Tbx3*-mediated repression through VCS conditional haploinsufficiency allows better toleration of *Tbx5* VCS haploinsufficiency. Conversely, excess *Tbx3*-mediated repression through overexpression results in disruption of the fast-conduction gene network despite normal levels of *Tbx5*. Based on these data the authors proposed a model in which repressive functions of *Tbx3* drive the adoption of conduction system fate, followed by segregation into a fast-conducting VCS and slow-conduction AVN through modulation of the *Tbx5/Tbx3* ratio in these respective tissue compartments.The question motivating the present work is: If *Tbx5/Tbx3* ratio is important for slow versus fast VCS identity, what happens when both genes are completely deleted from the VCS? Is conduction system identity completely lost without both factors and if so, does the VCS network transform into a working myocardium-like state? To address this question, the authors have generated a novel mouse line in which both *Tbx5* and *Tbx3* are floxed on the same allele, allowing complete conditional deletion of both factors using the VCS-specific *MinK-CreERT2* line, convincingly validated in previous work. The goal is to use these double conditional knockout mice to further explore the model of *Tbx3/Tbx5* co-dependent gene networks and VCS patterning. First, the authors demonstrate that the double conditional knockout allele results in the expected loss of *Tbx3* and *Tbx5* specifically in the VCS when crossed with *Mink-CreERT2* and induced with tamoxifen. The double conditional knockout also results in premature mortality. Detailed electrophysiological phenotyping demonstrated prolonged PR and QRS intervals, inducible ventricular tachycardia, and evidence of abnormal impulse propagation along the septal aspect of the right ventricle. In addition, the mutants exhibit downregulation of VCS genes responsible for both fast conduction AND slow conduction phenotypes with upregulation of 2 working myocardial genes including connexin-43. The authors conclude that loss of both *Tbx3* and *Tbx5* results in "reversion" or "transformation" of the VCS network to a working myocardial phenotype, which they further claim is a prediction of their model and establishes that *Tbx3* and *Tbx5* "coordinate" transcriptional control of VCS identity.

We appreciate Reviewer #2’s detailed summary of the study’s aims, methodologies, and findings, as well as their thoughtful suggestions for further analysis. We are grateful for their recognition of our genetic model’s novelty and robustness.

Overall Appraisal:As noted above, the present study does not further explore the *Tbx5/Tbx3* ratio concept since both genes are completely knocked out in the VCS. Instead, the main claims are that the absence of both factors results in a transcriptional shift of conduction tissue towards a working myocardial phenotype, and that this shift indicates that *Tbx5* and *Tbx3* "coordinate" to control VCS identity and function.

We agree with this reviewer’s assessment of the assertions in our manuscript. The novel combined *Tbx5/Tbx3* double mutant model does not further explore the TBX5/TBX3 ratio concept, which we previously examined in detail (1). Instead, as the Reviewer notes, this manuscript focuses on testing a model that the coordinated activity of *Tbx3* and *Tbx5* defines specialized ventricular conduction identity.

(1) Burnicka-Turek O, Broman MT, Steimle JD, Boukens BJ, Petrenko NB, Ikegami K, Nadadur RD, Qiao Y, Arnolds DE, Yang XH, Patel VV, Nobrega MA, Efimov IR, Moskowitz IP (2020) Transcriptional Patterning of the Ventricular Cardiac Conduction System. Circulation Research 127:e94-e106. doi:10.1161/CIRCRESAHA.118.314460.

Strengths:(1) Successful generation of a novel *Tbx3-Tbx5* double conditional mouse model.(2) Successful VCS-specific deletion of *Tbx3* and *Tbx5* using a VCS-specific inducible Cre driver line.(3) Well-powered and convincing assessments of mortality and physiological phenotypes. (4) Isolation of genetically modified VCS cells using flow.

We thank *Reviewer #2* for acknowledging the listed strengths of our study.

Weaknesses:(1) In general, the data is consistent with a long-standing and well-supported model in which *Tbx3* represses working myocardial genes and *Tbx5* activates the expression of VCS genes, which seem like distinct roles in VCS patterning. However, the authors move between different descriptions of the functional relationship and epistatic relationship between these factors, including terms like "cooperative", "coordinated", and "distinct" at various points. In a similar vein, sometimes terms like "reversion" are used to describe how VCS cells change after *Tbx3/Tbx5* conditional knockout, and other times "transcriptional shift" and at other times "reprogramming". But these are all different concepts. The lack of a clear and consistent terminology for describing the phenomena observed makes the overarching claims of the manuscript more difficult to evaluate.

We discriminate prior work on the “long-standing and well-supported model’ supported by investigation of the role of *Tbx5* and *Tbx3* independently from this work examining the coordinated role of *Tbx5* and *Tbx3*. Prior work demonstrated that *Tbx3* represses working myocardial genes and *Tbx5* activates expression of VCS genes, consistent with the reviewer’s suggestion of their distinct roles in VCS patterning. However, the current study uniquely evaluates the combined role of *Tbx3* and *Tbx5* in distinguishing specialized conduction identify from working myocardium, for the first time.

We appreciate Reviewer #2’s feedback regarding the need for consistent terminology when describing the impact of the double *Tbx3* and *Tbx5* mutant. We will edit the manuscript to replace terms like “reversion” with “transcriptional shift” or “transformation” when describing the observed phenotype, and we will use “coordination” to describe the combined role of *Tbx5* and *Tbx3* in maintaining VCS-specific identity.

(2) A more direct quantitative comparison of *Tbx5* Adult VCS KO with *Tbx5/Tbx3* Adult VCS double KO would be helpful to ascertain whether deletion of *Tbx3* on top of *Tbx5* deletion changes the underlying phenotype in some discernable way beyond mRNA expression of a few genes. Superficially, the phenotypes look quite similar at the EKG and arrhythmia inducibility level and no optical mapping data from a single *Tbx5* KO is presented for comparison to the double KO.

We thank Reviewer #2 for the suggestions that a direct comparison between *Tbx5* single conditional knockout and *Tbx3/Tbx5* double conditional knockout models may help isolate the specific contribution of *Tbx3* deletion in addition to *Tbx5* deletion.

Previous studies have assessed the effect of single *Tbx5* CKO in the VCS of murine hearts (1, 3, 5). Arnolds *et al.* demonstrated that the removal of *Tbx5* from the adult ventricular conduction system results in VCS slowing, including prolonged PR and QRS intervals, prolongation of the His duration and His-ventricular (HV) interval (3).

Furthermore, Burnicka-Turek *et al.* demonstrated that the single conditional knockout of *Tbx5* in the adult VCS caused a shift toward a pacemaker cell state, with ectopic beats and inappropriate automaticity (1). Whole-cell patch clamping of VCS-specific *Tbx5* deficient cells revealed action potentials characterized by a slower upstroke (phase 0), prolonged plateau (phase 2), delayed repolarization (phase 3), and enhanced phase 4 depolarization - features characteristic of nodal action potentials rather than typical VCS action potentials (3). These observations were interpreted as uncovering nodal potential of the VCS in the absence of *Tbx5*. Based on the role of Tbx3 in CCS specification (2), we hypothesized that the nodal state of the VCS uncovered in the absence of *Tbx5* was enabled by maintained *Tbx3* expression. This motivated us to generate the double *Tbx5*

*/ Tbx3* knockout model to examine the state of the VCS in the absence of both T-box TFs. In the current study, we demonstrate that the VCS-specific deletion of *Tbx3* and *Tbx5* results in the loss of fast electrical impulse propagation in the VCS, similar to that observed in the single *Tbx5* mutant. However, unlike the *Tbx5* single mutant, the *Tbx3/Tbx5* double deletion does not cause a gain of pacemaker cell state in the VCS. Instead, the physiological data suggests a transition toward non-conduction working myocardial physiology. This conclusion is supported by the presence of only a single upstroke in the optical action potential (OAP) recorded from the His bundle region and VCS cells in *Tbx3/Tbx5* double conditional knockout mice. The electrical properties of VCS cells in the double knockout are functionally indistinguishable from those of ventricular working myocardial cells. As a result, ventricular impulse propagation is significantly slowed, resembling activation through exogenous pacing rather than the rapid conduction typically associated with the VCS. We will edit the text of the manuscript to more carefully distinguish the observations between these models, as suggested.

(1) Burnicka-Turek O, Broman MT, Steimle JD, Boukens BJ, Petrenko NB, Ikegami K, Nadadur RD, Qiao Y, Arnolds DE, Yang XH, Patel VV, Nobrega MA, Efimov IR, Moskowitz IP (2020) Transcriptional Patterning of the Ventricular Cardiac Conduction System. Circulation Research 127:e94-e106. doi:10.1161/CIRCRESAHA.118.314460.

(2) Mohan RA, Bosada FM, van Weerd JH, van Duijvenboden K, Wang J, Mommersteeg MTM, Hooijkaas IB, Wakker V, de Gier-de Vries C, Coronel R, Boink GJJ, Bakkers J, Barnett P, Boukens BJ, Christoffels VM (2020) T-box transcription factor 3 governs a transcriptional program for the function of the mouse atrioventricular conduction system. Proc Natl Acad Sci U S A. 117:18617-18626. doi: 10.1073/pnas.1919379117.

(3) Arnolds DE, Liu F, Fahrenbach JP, Kim GH, Schillinger KJ, Smemo S, McNally EM, Nobrega MA, Patel VV, Moskowitz IP (2012) TBX5 drives Scn5a expression to regulate cardiac conduction system function. The Journal of Clinical Investigation 122:2509–2518. doi: 10.1172/JCI62617.

(4) Frank DU, Carter KL, Thomas KR, Burr RM, Bakker ML, Coetzee WA, Tristani-Firouzi M, Bamshad MJ, Christoffels VM, Moon AM (2012) Lethal arrhythmias in Tbx3-deficient mice reveal extreme dosage sensitivity of cardiac conduction system function and homeostasis. Proc Natl Acad Sci U S A. 109:E154-63. doi: 10.1073/pnas.1115165109.

(5) Moskowitz IP, Pizard A, Patel VV, Bruneau BG, Kim JB, Kupershmidt S, Roden D, Berul CI, Seidman CE, Seidman JG (2004) The T-Box transcription factor Tbx5 is required for the patterning and maturation of the murine cardiac conduction system. Development 131:4107-4116. doi: 10.1242/dev.01265. PMID: 15289437.

(3) The authors claim that double knockout VCS cells transform to working myocardial fate, but there is no comparison of gene expression levels between actual working myocardial cells and the *Tbx3/Tbx5* DKO VCS cells so it's hard to know if the data reflect an actual cell state change or a more non-specific phenomenon with global dysregulation of gene expression or perhaps dedifferentiation. I understand that the upregulation of *Gja1* and *Smpx* is intended to address this, but it's only two genes and it seems relevant to understand their degree of expression relative to actual working myocardium. In addition, the gene panel is somewhat limited and does not include other key transcriptional regulators in the VCS such as *Irx3* and *Nkx2-5*. RNA-seq in these populations would provide a clearer comparison among the groups.Andthe main claims are that the absence of both factors results in a transcriptional shift of conduction tissue towards a working myocardial phenotype, and that this shift indicates that *Tbx5* and *Tbx3* "coordinate" to control VCS identity and function. However, only limited data are presented to support the claim of transcriptional reprogramming since the knockout cells are not directly compared to working myocardial cells at the transcriptional level and only a small number of key genes are assessed (versus genome-wide assessment).

We appreciate Reviewer #2’s suggestion to expand the gene expression analysis in *Tbx3/Tbx5*-deficient VCS cells by including other specific genes and comparisons with “native”/actual working ventricular myocardial cells and broadening the gene panel. In this study, we evaluated core cardiac conduction system markers, revealing a loss of conduction system-specific gene expression in the double mutant VCS. Furthermore, we evaluated key working myocardial markers normally excluded from the conduction system, *Gja1* and *Smpx*, revealing a shift towards a working myocardial state in the double mutant VCS (Figure 4). We agree that a more comprehensive analysis, such as transcriptome-wide approaches, would offer greater clarity on the extent and specificity of the observed shift from conduction to non-conduction identity. These approaches are appropriate directions for future studies.

(4) From the optical mapping data, it is difficult to distinguish between the presence of (a) a focal proximal right bundle branch block due to dysregulation of gene expression in the VCS but overall preservation of the right bundle and its distal ramifications; from (b) actual loss of the VCS with reversion of VCS cells to a working myocardial fate. Related to this, the authors claim that this experiment allows for direct visualization of His bundle activation, but can the authors confirm or provide evidence that the tissue penetration of their imaging modality allows for imaging of a deep structure like the AV bundle as opposed to the right bundle branch which is more superficial? Does the timing of the separation of the sharp deflection from the subsequent local activation suggest visualization of more distal components of the VCS rather than the AV bundle itself? Additional clarification would be helpful.AndIn addition, the optical mapping dataset is incomplete and has alternative interpretations that are not excluded or thoroughly discussed.

We agree with Reviewer #2 that the resolution of the optical mapping experiment may be insufficient to precisely localize the conduction block due to the limited signal strength from the VCS. It is possible that the region defined as the His Bundle also includes portions of the right bundle branch. Our control mice show VCS OAP upstrokes consistent with those reported by Tamaddon *et al.* (2000) using Di-4-ANEPPS (1). We appreciate the Reviewer’s attention to alternative interpretations, and we will incorporate these caveats into the manuscript text.

(1) Tamaddon HS, Vaidya D, Simon AM, Paul DL, Jalife J, Morley GE (2000) Highresolution optical mapping of the right bundle branch in connexin40 knockout mice reveals slow conduction in the specialized conduction system. Circulation Research 87:929-36. doi: 10.1161/01.res.87.10.929.

Impact:The present study contributes a novel and elegantly constructed mouse model to the field. The data presented generally corroborate existing models of transcriptional regulation in the VCS but do not, as presented, constitute a decisive advance.AndIn sum, while this study adds an elegantly constructed genetic model to the field, the data presented fit well within the existing paradigm of established functions of *Tbx3* and *Tbx5* in the VCS and in that sense do not decisively advance the field. Moreover, the authors' claims about the implications of the data are not always strongly supported by the data presented and do not fully explore alternative possibilities.

We appreciate Reviewer # 2’s acknowledgment of the elegance and novelty of the mouse model we generated. However, we respectfully disagree with their assessment that this work merely corroborates existing models without providing a decisive advance. Previous studies have investigated single *Tbx5* or *Tbx3* gene knockouts in-depth and established the T-box ratio model for distinguishing fast VCS from slow nodal conduction identity (1) that the reviewer alludes to in earlier comments. In contrast, this study aimed to explore a different model, that the combined effects of *Tbx5* and *Tbx3* distinguish adult VCS identity from non-conduction working myocardium. The coordinated *Tbx3* and *Tbx5* role in conduction system identify remained untested due to the lack of a mouse model that allowed their simultaneous removal. The very model the reviewer recognizes as “novel and elegantly constructed” has allowed the examination of the coordinated role of *Tbx5* and *Tbx3* for the first time. While we acknowledge the opportunity for additional depth of investigation of this model in future studies, the data we present provides consistent experimental support for the coordinated requirement of both *Tbx5* and *Tbx3* for ventricular cardiac conduction system identity.

(1) Burnicka-Turek O, Broman MT, Steimle JD, Boukens BJ, Petrenko NB, Ikegami K, Nadadur RD, Qiao Y, Arnolds DE, Yang XH, Patel VV, Nobrega MA, Efimov IR, Moskowitz IP (2020) Transcriptional Patterning of the Ventricular Cardiac Conduction System. *Circulation Research* 127:e94-e106. doi:10.1161/CIRCRESAHA.118.314460.

**Reviewer #3 (Public review):**
Summary:In the study presented by Burnicka-Turek *et al.,* the authors generated for the first time a mouse model to cause the combined conditional deletion of *Tbx3* and *Tbx5* genes. This has been impossible to achieve to date due to the proximity of these genes in chromosome 5, preventing the generation of loss of function strategies to delete simultaneously both genes. It is known that both *Tbx3* and *Tbx5* are required for the development of the cardiac conduction system by transcription factor-specific but also overlapping roles as seen in the common and diverse cardiac defects found in patients with mutations for these genes. After validating the deletion efficiency and specificity of the line, the authors characterized the cardiac phenotype associated with the cardiac conduction system (CCS)-specific combined deletion of T_bx5_ and *Tbx3* in the adult by inducing the activation of the CCS-specific tamoxifen-inducible Cre recombination (*MinKcreERT*) at 6 weeks after birth. Their analysis of 8-9-week-old animals did not identify any major morphological cardiac defects. However, the authors found conduction defects including prolonged PR and QTR intervals and ventricular tachycardia causing the death of the double mutants, which do not survive more than 3 months after tamoxifen induction. Molecular and optical mapping analysis of the ventricular conduction system (VCS) of these mutants concluded that, in the absence of *Tbx5* and *Tbx3* function, the cells forming the ventricular conduction system (VCS) become working myocardium and lose the specific contractile features characterizing VCS cells. Altogether, the study identified the critical combined role of *Tbx3* and *Tbx5* in the maintenance of the VCS in adulthood.Strengths:The study generated a new animal model to study the combined deletion of *Tbx5* and *Tbx3* in the cardiac conduction system. This unique model has provided the authors with the perfect tool to answer their biological questions. The study includes top-class methodologies to assess the functional defects present in the different mutants analyzed, and gathered very robust functional data on the conduction defects present in these mutants. They also applied optical action potential (OAP) methods to demonstrate the loss of conduction action potential and the acquisition of working myocardium action potentials in the affected cells because of *Tbx5/Tbx3* loss of function. The study used simpler molecular and morphological analysis to demonstrate that there are no major morphological defects in these mutants and that indeed, the conduction defects found are due to the acquisition of working myocardium features by the VCS cells. Altogether, this study identified the critical role of these transcription factors in the maintenance of the VCS in the adult heart.

We appreciate the Reviewer’s comments regarding the originality and utility of our model and the strengths of our methodological approach. The Reviewer’s appreciation of the molecular and morphological analyses as well as their constructive feedback is highly valuable.

Weaknesses:In the opinion of this reviewer, the weakness in the study lies in the morphological and molecular characterization. The morphological analysis simply described the absence of general cardiac defects in the adult heart, however, whether the CCS tissues are present or not was not investigated. Lineage tracing analysis using the reporter lines included in the crosses described in the study will determine if there are changes in CCS tissue composition in the different mutants studied. Similarly, combining this reporter analysis with the molecular markers found to be dysregulated by qPCR and western blot, will demonstrate that indeed the cells that were specified as VCS in the adult heart, become working myocardium in the absence of *Tbx3* and *Tbx5* function.

We appreciate the reviewer’s concern regarding the morphology of the cardiac conduction system in the *Tbx3/Tbx5* double conditional knockout model. We did not observe any structural abnormalities, as the Reviewer notes. We agree with their suggestion for using Genetic Inducible Fate Mapping to mark cardiac conduction cells expressing *MinKCre*. In fact, we utilized this approach to isolate VCS cells for transcriptional profiling. Specifically, we combined the tamoxifen-inducible *MinKCreERT* allele with the Cre-dependent *R26Eyfp* reporter allele to label *MinKCre*-expressing cells in both control VCS and VCS-specific double *Tbx3*/*Tbx5* knockouts. EYFP-positive cells were isolated for transcriptional studies, ensuring that our analysis exclusively targeted conduction system-lineage marked cells. The ability to isolate *MinKCre*-marked cells from both controls and Tbx5/Tbx3 double mutants indicates that VCS cells persisted in the double knockout. Nonetheless, the suggestion for *in-vivo* marking by Genetic Inducible

Fate Mapping and morphologic analysis is a valuable recommendation for future studies.

**Reviewer #1 (Recommendations for the authors):**
In a heroic effort, Ozanna Burnicka-Turek et al. have made and investigated conduction system-specific *Tbx3-Tbx5* deficient mice and investigated their cardiac phenotype. Perhaps according to expectations, given the body of literature on the function of the two T-box transcription factors in the heart/conduction system, the cardiomyocytes of the ventricular conduction system seemed to convert to "ordinary" ventricular working myocytes. As a consequence, loss of VCS-specific conduction system propagation was observed in the compound KO mice, associated with PR and QRS prolongation and elevated susceptibility to ventricular tachycardia.Previous work suggested the prediction that VCS-specific genetic ablation of both the TBX3 and TBX5 would transform fast-conducting adult VCS into cells resembling working myocardium, eliminating specialized CCS fate. The current study suggests that this prediction is at least to some extent accurate.

We appreciate Reviewer #1’s summary and recognition of our study. As the review notes, the simultaneous deletion of *Tbx3* and *Tbx5* in the mature ventricular conduction system (VCS) suggests a conversion of VCS to "ordinary" ventricular working myocytes. To our knowledge, this represents a novel observation and experimental model that uniquely captures the combined roles of these essential T-box transcription factors. We believe that this model offers a valuable platform for further investigation into the transcriptional mechanisms underlying conduction system specialization.

(1) The huge effort made to generate the DKO model contrasts with the limited efforts made to study the mechanism. Conditional deficiency of *Tbx3* and *Tbx5* creates an artificial situation that is useful for addressing fundamental mechanistic questions. The authors provide a rather superficial analysis of the changes in the VCS upon deletion of these two critically important factors and do not provide really novel insights into their requirement/function in the VCS gene regulatory network and epigenetic state. So to what extent do VCS cardiomyocytes (CMs) from *Tbx3/5* DKO mice resemble "simple" working myocardium? To what extent do these cells acquire the working myocardial (epigenetic) state, do these cells have an epigenetic memory of the *Tbx3/Tbx5+* history, is the enhancer usage between the modified VCS CMs and the working CMs similar or not, etc.? The assumption that the authors' data indicate that the DKO VCS CMs simply acquire a ventricular working "fate" is unlikely. Following this reasoning, the reverse experiment to induce *Tbx3* and *Tbx5* expression in working CMs would result in complete conversion to VCS CMs, which is also unlikely.To answer such questions, transcriptomic and epigenetic state analysis, electrophysiologic analysis (e.g. patch-clamp), cell/subcellular level analysis, etc. would be required, as well as a comparison of the changed state of the DKO VCS CMs to that of working CMs.

This initial study focused on generating the *Tbx3:Tbx5* double-conditional knockout model and characterizing the resulting physiological and molecular changes within the VCS. We analyzed transcriptomic markers of fast conduction (VCS), slow conduction (nodal), and non-conduction (working myocardium). Additionally, we applied optical mapping to evaluate the physiological consequences of the double knockout, which allowed a calculated AP of the VCS to be generated. We agree that a more in-depth mechanistic investigation of the VCS transformation upon *Tbx3/Tbx5* deletion by transcriptomic or cellular electrophysiology could provide a deeper understanding of the precise transcriptional/epigenetic state of the VCS in the double knockout and clarify whether there is a partial or complete conversion of VCS cells to a simple working myocardial phenotype. The suggestions by the reviewer will be considered for future studies.

(2) *Tbx3* stimulates BMP-TGFb signaling (e.g. positive loop between *Tbx3-Bmp2*), which in turn stimulates EMT and modulates the behavior of endocardial and mesenchymal cells. Did the authors investigate the impact of *Tbx3/5* DKO on non-CM cells in and around the VCS? (see also comment 1). The insulation of the AVB for example could be a *Tbx3/5* non cell autonomous target.

We appreciate the Reviewer’s suggestion to examine the impact of *Tbx3/Tbx5* deletion on non-CM cells surrounding the VCS. While this is an intriguing avenue for future exploration, it falls outside the scope of the current study, which focused on the cardiomyocyte-specific roles of *Tbx3* and *Tbx5* in maintaining adult VCS identity.

(3) The *MinK-Cre* line used (from the Moskowitz lab) also recombines in the AVN (Arnolds *et al* 2011). The authors do not mention changes in the AVN, and systematically call the line VCS specific (which refers to the AVB, BB, PVCS I assume). This could also impact the PR interval. Please address.

The *MinK-Cre* line recombines in the atrioventricular bundle (AVB) and bundle branches (BB). It recombines in cardiomyocytes adjacent to the atrioventricular node (AVN). We previously interpreted these cells as the penetrating portion of the His bundle into the AVN. This line does not recombine in the vast majority, if any, physiologic nodal cells. We also assessed nodal conduction parameters by invasive electrophysiologic (EP) studies. Our data showed that non-VCS parameters, including sinus node recovery time, AV node recovery time, and atrial and ventricular effective refractory periods, remained within normal ranges in *Tbx3:Tbx5*-deficient mice (please see Figure 2I). These findings indicate that AVN function is preserved in the VCS-specific double knockout, reinforcing the specificity of the observed conduction defects to the ventricular conduction system.

(4) Did the authors also investigate the electrophysiological changes in the (EGFP+) DKO VCS CMs? Would these resemble the properties of ventricular working CMs, or would they still show some VCS properties? (see also comment 1).

We performed electrophysiologic analysis of the double knockout by optical mapping. Optical mapping provides tissue-level resolution, capturing the functional behavior of clusters of thousands of cells simultaneously, rather than individual cells. While this technique does not achieve single-cell resolution, it allows for a comprehensive assessment of electrophysiological changes across the VCS region. Single cell electrophysiology is a good idea for future studies.

(5) Throughout the manuscript, the authors use "patterning" and "fate", which are applicable to development and differentiation, not to the situation where a gene is removed from fully differentiated cells in an adult organism resulting in a change of these cells. Perhaps more appropriate are "state" change and the requirement for "homeostasis/maintenance" of state.

We appreciate the Reviewer’s concern regarding the terminology used to describe changes in VCS cell identity. To ensure precision and uniformity, we replaced terms such as “fate” and “patterning” with “state” or “maintenance” to reflect the shift in cellular characteristics in a fully differentiated adult tissue context.

Minor:(1) Please provide all data points in bar graphs.

We have incorporated individual data points into the bar graphs as suggested, ensuring enhanced transparency and clarity in the data presentation.

“(2) Formally, gene expression levels between samples are not normally distributed. The Welch t-test used here assumes a normal distribution. Therefore, nonparametric tests should be used.

We appreciate Reviewer #1’s consideration of the appropriate statistical approach to the qPCR data and clarify our statistical approach here. Normality within each experimental group was assessed using the Shapiro-Wilk test. Between-group comparisons were conducted using Welch t-test, and multiple comparisons were corrected using the Benjamini & Hochberg method to control the false discovery rate (FDR) (71). If a significant difference was detected between two groups (t-test FDR < 0.05) but normality was rejected in any of the compared groups (Shapiro-Wilk P < 0.05), a non-parametric Wilcoxon rank-sum test was used for verification. A significant group-mean difference was confirmed at one-tailed Wilcoxon P≤0.05 (detailed in Supplementary Data Set I). Furthermore, we have updated the qRT-PCR information in each figure and their respective legends as follows. Statistical analysis was performed using R version 4.2.0. We have included a new Supplementary Data Set I, detailing the statistical analysis of qRT-PCR data. Additionally, we have revised the Methods/Statistics section to detail the applied statistical analysis.

(3) Some of the panels of figures are tiny and cannot be evaluated. For example, in Figure 1B the actual data (expression of *Tbx3/5*) is impossible to see.

We appreciate the Reviewer’s observation and have revised the figures to improve visual clarity and ensure that the presented data are easily interpretable by readers.

**Reviewer #2 (Recommendations for the authors):**
Additional Experiments, Data, Analysis:(1) Comparisons between both single knockouts and double knockouts at the phenotypic level are needed. In some instances, the data is shown (e.g., mortality and EKG) but direct statistical comparison is not performed. In other instances (optical mapping and gene expression), data with single knockouts are not shown. If combined VCS *Tbx3/Tbx5* deletion does not change the phenotype of the VCS *Tbx5* single deletion, this should be explicitly stated and discussed.

We appreciate Reviewer #2’s suggestion to compare the phenotypic outcomes of the *Tbx3* and *Tbx5* single conditional knockout models with those observed in *Tbx3/Tbx5* double conditional knockout model. We have expanded the discussion section of our manuscript to incorporate a more detailed comparison between the double *Tbx3/Tbx5* model and the single *Tbx5* and *Tbx3* models [1-5], highlighting the distinct phenotypic outcomes of the single and double knockouts.

(1) Burnicka-Turek O, Broman MT, Steimle JD, Boukens BJ, Petrenko NB, Ikegami K, Nadadur RD, Qiao Y, Arnolds DE, Yang XH, Patel VV, Nobrega MA, Efimov IR, Moskowitz IP (2020) Transcriptional Patterning of the Ventricular Cardiac Conduction System. Circulation Research 127:e94-e106. doi:10.1161/CIRCRESAHA.118.314460.

(2) Mohan RA, Bosada FM, van Weerd JH, van Duijvenboden K, Wang J, Mommersteeg MTM, Hooijkaas IB, Wakker V, de Gier-de Vries C, Coronel R, Boink GJJ, Bakkers J, Barnett P, Boukens BJ, Christoffels VM (2020) T-box transcription factor 3 governs a transcriptional program for the function of the mouse atrioventricular conduction system. Proc Natl Acad Sci U S A. 117:18617-18626. doi: 10.1073/pnas.1919379117.

(3) Arnolds DE, Liu F, Fahrenbach JP, Kim GH, Schillinger KJ, Smemo S, McNally EM, Nobrega MA, Patel VV, Moskowitz IP (2012) TBX5 drives Scn5a expression to regulate cardiac conduction system function. The Journal of Clinical Investigation 122:2509–2518. doi: 10.1172/JCI62617.

(4) Frank DU, Carter KL, Thomas KR, Burr RM, Bakker ML, Coetzee WA, Tristani-Firouzi M, Bamshad MJ, Christoffels VM, Moon AM (2012) Lethal arrhythmias in Tbx3-deficient mice reveal extreme dosage sensitivity of cardiac conduction system function and homeostasis. Proc Natl Acad Sci U S A. 109:E154-63. doi: 10.1073/pnas.1115165109. [5] Moskowitz IP, Pizard A, Patel VV, Bruneau BG, Kim JB, Kupershmidt S, Roden D, Berul CI, Seidman CE, Seidman JG (2004) The T-Box transcription factor Tbx5 is required for the patterning and maturation of the murine cardiac conduction system. Development 131:4107-4116. doi: 10.1242/dev.01265.

(2) Genome-wide expression analysis including working myocardium would provide stronger evidence for interconversion of cell states. Ideally, this would include single knockouts.

We agree that a genome-wide expression analysis, including a direct comparison with working myocardium, would provide more comprehensive insights into cell state transitions in *Tbx3:Tbx5*-deficient VCS cells. Additionally, incorporating single knockout models into such analyses would further clarify the distinct and cooperative contributions of *Tbx3* and *Tbx5* to maintaining VCS identity. This is a good suggestion for future studies.

(3) This may not be essential to support the authors' claims, but the addition of epigenetic data from single and double KO VCS using ATAC-seq (which can be performed with relatively small numbers of cells) could provide stronger evidence for cell state changes of the kind hypothesized by the authors.

We agree that epigenetic data such as ATAC-seq would complement transcriptional analyses and provide insight into chromatin states that underlie the observed cellular reprogramming. This is a good suggestion for follow-up studies to further characterize the molecular state of *Tbx3:Tbx5*-deficient VCS cells.

(4) Additional clarification of the optical mapping experiments to exclude alternative interpretations like focal right bundle branch block and to include single knockouts for comparison - if the *Tbx5* single KO looks the same as the double KO that would be very important to know and would directly affect interpretation of the experiment.

Right septal optical mapping preparation involved removing the right ventricular free wall to directly image the right ventricular septum, which contains the VCS. In a healthy mouse, there are two peak components of the optical action potential upstroke, the first peak due to the activation of the VCS and the second due to the activation of the ventricular cardiomyocytes. Importantly, in *Tbx3:Tbx5* double-conditional knockout mice, the first peak was absent, rather than delayed, indicating loss of fast conduction through the VCS. This absence suggests a shift in VCS cells toward a ventricular working myocardial phenotype, rather than a regional conduction block or delayed propagation through a structurally intact VCS.

Previous studies from our group have extensively characterized the effect of single *Tbx5* knockout on the VCS in murine hearts [1, 2, 3]. Arnolds et al. demonstrated that VCSspecific *Tbx5*-deficiency results in significant slowing of VCS conduction, evidenced by prolonged PR and QRS intervals, along with lengthening of the atrio-Hisian interval, His duration, and Hisioventricular interval [1]. Although both single *Tbx5* knockout and *Tbx3:Tbx5* double knockout mice exhibit slowing of ventricular conduction system, our optical mapping studies reveal distinct differences in their electrophysiological phenotypes. Burnicka-Turek *et al.* showed that the single knockout of *Tbx5* in the VCS leads to a shift toward a pacemaker cell state, evidenced by ectopic beats originating in the ventricles and inappropriate automaticity [3]. During spontaneous beats, electrical impulses were retrogradely activated, propagating from the ventricles to the atria [3]. Whole-cell patch clamping recordings confirmed that *Tbx5*-deficient VCS cells displayed action potentials resembling pacemaker cells, characterized by slower upstroke (phase 0), prolonged plateau (phase 2), delayed repolarization (phase 3), and enhanced phase 4 depolarization [3]. In contrast, our current study on VCS-specific *Tbx3:Tbx5* double knockout demonstrates a loss of the VCS-specific fast conduction propagation. Optical mapping demonstrated the absence of the initial upstroke corresponding to VCS activation in the His bundle region, indicating a shift in the VCS cells toward a ventricular working myocardium state. This loss of fast conduction properties highlights a fundamental distinction between single and double knockouts, suggesting that both *Tbx3* and *Tbx5* are required to maintain VCS identity and function.

(1) D. E. Arnolds et al., “TBX5 drives Scn5a expression to regulate cardiac conduction system function,” J. Clin. Invest., vol. 122, no. 7, pp. 2509–2518, Jul. 2012, doi: 10.1172/JCI62617.

(2) Moskowitz, I.P., Pizard, A., Patel, V.V., Bruneau, B.G., Kim, J.B., Kupershmidt, S., Roden, D., Berul, C.I., Seidman, C.E., Seidman, J.G. (2004) The T-Box transcription factor Tbx5 is required for the patterning and maturation of the murine cardiac conduction system. Development 131(16):4107-4116.

(3) Burnicka-Turek, O., Broman, M.T., Steimle, J.D., Boukens, B.J., Peterenko, N.B, Ikegami, K., Nadadur, R.D., Qiao, Y., Arnolds, D.E., Yang, X.H., Patel, V.V., Nobrega, M.A., Efimov, I.R., Moskowitz, I.P. (2020) Transcriptional Patterning of the Ventricular Cardiac Conduction System. Circ Res. 127(3):e94-e106.

Methods:(1) Additional methods on FACS are required. The methods section references a paper from 2004 (reference 67) that describes the flow sorting of embryonic cardiomyocytes. However, flow cytometric isolation of intact adult cardiomyocytes, which the authors describe in the present work, is a distinct technique and generally requires special equipment. These need to be described in more detail to be fully replicable.

We thank Reviewer #2 for highlighting the need to provide additional details regarding our flow cytometric isolation of adult VCS cardiomyocytes. While we referenced earlier methods, we agree that isolating adult cardiomyocytes requires specialized approaches. Therefore, we revised the Methods section to include a detailed description of the equipment, procedures, and adaptations specific to isolating intact adult VCS cells to ensure full replicability.

Minor Corrections:

(1) Figure 1D. Please add a statistical test for mortality between the double conditional KO and the *Tbx5* conditional KO.

We have revised Figure 1D to include the statistical test comparing mortality between the *Tbx3:Tbx5* double conditional knockout and the *Tbx5* conditional knockout cohorts.

(2) Figure 2A, 2I, 3A: Please include all individual data points not just a bar graph with error bars.

We have added all individual data points to the bar graphs as recommended, enhancing the transparency and clarity of the data presentation.

(3) Figure 2A: Please consider separate graphs for PR and QRS with appropriately scaled Y-axis so differences are easier to see.

We appreciate Reviewer #2’s suggestion and fully agree with it. As a result, we have revised Figure 2A to include separate graphs for PR and QRS intervals, each with appropriately scaled Y-axes. This adjustment enhanced both the readability and the clarity of the observed differences.

(4) Figure 3 G-K: The figure would be easier to interpret for the reader if genotypes were shown in the figure not just in the legend.

We agree with Reviewer #2’s suggestion and have revised Figure 3 accordingly by adding genotype labels directly to the histological sections in Panels G-K. This update improves clarity, making the data easier for readers to interpret without needing to refer to the figure legend.

(5) Figure 4A, C: Are vertical axes mislabeled? They say, "CON VCS and TBX5OE VCS". Please double-check axis labels and data on the graph.

We appreciate the Reviewer bringing the mislabeling of the vertical axis in Figure 4 to our attention. We have corrected the labeling errors and ensured consistency between the graph and the underlying data.

(6) Legend to Supplementary Figure 6. Says "*Tbx3:Tbx3*" instead of "*Tbx3:Tbx5*".

We thank Reviewer #2 for pointing out the typo. It has been corrected to: “Supplementary Figure 6. *Tbx3:Tbx5* double-conditional knockout mice exhibit QRS prolongation”.

(7) Discussion. The authors write, "In *Tbx3:Tbx5* double VCS knockout, we observed repression of fast VCS markers and also repression of Pan-CCS markers transcribed throughout the entire CCS." The term 'repression' has a specific connotation with transcription regulators that is likely not intended in this context so perhaps 'reduced expression' would be better here?

We agree with Reviewer #2 and have replaced “repression” with “reduced expression” throughout the text (look below for references).

“In the *Tbx3:Tbx5* double VCS knockout, we observed a reduction in the expression of both fast VCS markers and Pan-CCS markers transcribed throughout the entire CCS.”

(8) Discussion, the authors write, "This study combined with prior literature (1, 7, 11, 15, 26, 53, 54) indicates that the presence of both *Tbx3* and *Tbx5* is necessary for the specification of the adult VCS (Figure 7)." Since this work presents data from an adult conditional deletion, it's not clear how it informs our understanding of the specification, which occurs during development. Perhaps "maintenance of VCS fate" would be more appropriate here?

We agree with Reviewer #2 that the term “maintenance of VCS fate” is more appropriate in the context of our study. Accordingly, we have updated the text to reflect this terminology.

**Reviewer #3 (Recommendations for the authors):**
(1) Figure 2B: It is hard to see the IF images. What is the cardiac structure studied? Maybe a dashed line and a label to define the region and the structure represented will help. As the authors have described that the crosses used contain a reporter allele (*R26-EYFP*), a clearer way to show these results would be to include images of the linage traced cells with the reporter, not only to identify the CCS structure analyzed, but also to demonstrate that the deletion is specific to the *MinK-creERT* expression in the CCS.

We appreciate the Reviewer’s suggestion to improve the clarity of Figure 2B by delineating the cardiac structures analyzed. In response, we have added dashed lines and labels to highlight the regions of interest within the IF images. Unfortunately, we were unable to capture high-quality EYFP fluorescence images for these sections. However, to address this concern, we microdissected the region shown in the IF images and performed FACS to isolate EYFP-positive cells from this specific area. These sorted cells were subsequently used for qPCR analysis, which confirmed the presence of *Tbx3* and *Tbx5* in control samples and the successful deletion of both genes in the doubleconditional knockout samples (Figure 2C, middle panel). We believe this approach provides robust evidence for the specificity of the *MinK-CreERT* expression in the CCS and the efficiency of gene deletion in the targeted region.

(2) 3G-K: The authors describe the absence of morphological defects in the tissue sections of adult hearts from the different genotypes analyzed. Although this reviewer agrees that there seem to be no major defects in the general cardiac morphology of these animals, the higher magnification images suggest some tissue differences at the level of the AVN especially in the double HET, double HOMO, and the *Tbx3* HOMO. Is that due to the section plane used? If so, more appropriate and comparable sections must be provided. Again, as the crosses used by the authors contain a reporter allele (*R26-EYFP*), it is required that the authors show that the CCS cells, where deletions are induced, are still present in equivalent areas in the mutants and that they remain in similar numbers only failing to maintain their specification into CCS due to *Tbx3* and *Tbx5* loss of function.

This analysis will reinforce the authors' claims on the role of *Tbx5/Tbx3* in this process.

We thank the reviewer for their thorough assessment and thoughtful feedback on our histological analysis. The higher magnification images in Figure 3G-K do not specifically present the AVN. These sections primarily represent areas of the ventricular conduction system (VCS), particularly the His bundle and bundle branches, rather than the AVN itself. We do not believe that the observed morphological differences are related to AVN tissue, and there were no functional deficits attributable to the AVN in the double knockout. Furthermore, the *Mink-Cre* allele used in this study does not recombine in the ANV proper. We agree that confirming the presence of CCS cells in equivalent regions across different genotypes is crucial. Our approach using FACS-based isolation of EYFP-positive cells from the VCS, followed by qPCR analysis, provides evidence that these cells remain present in double conditional knockouts, although they fail to maintain their specialized gene expression profile. This reinforces our conclusion that *Tbx3* and *Tbx5* are essential for maintaining the molecular identity of CCS cells, rather than their physical presence.

(3) Figure 4: The authors performed molecular analysis by qPCR and WB in *Tbx5/Tbx3* double mutants to demonstrate that CCS cells lose the expression of CCS genes and express working myocardium genes. Could this be further demonstrated by ISH, HCR, or IF together with lineage tracing to provide evidence that these changes are located where the CCS tissues are in the control embryos? Analysis of 2 or 3 of these markers of each type on tissue sections would be enough.

We thank the Reviewer for their insightful suggestion regarding additional validation of our molecular findings through ISH, HCR, or IF combined with lineage tracing. However, we would like to clarify that the molecular analyses we performed by qPCR and WB were conducted on EYFP-positive cells that were specifically isolated from the ventricular conduction system (VCS) region of both control and double conditional knockout (dCKO) mice. These EYFP-positive cells were obtained through fluorescence-activated cell sorting (FACS), ensuring that our analyses were confined to the targeted VCS population. Alternate approaches are appropriate for future studies to investigate the precise genomic and molecular nature of the transformation observed in the double knockout.

(4) Discussion: in the discussion section the authors conclude that the combined role of *Tbx5/Tbx3* is critical for the specification of the adult VCS. However, as the *Tbx5/Tbx3* loss of function conditions are only induced in adult animals 6 weeks old, would it be more appropriate that their function is the maintenance of the VCS cell fate and that if not present these cells return to the working myocardium fate? If the authors believe that these genes are involved in the induction of VCS specification in adults, then they need to demonstrate that, before the loss of function induction at 6 weeks, these cells are not yet specified as adult VCS.

We appreciate the Reviewer’s clarification regarding terminology. We agree that our study focuses on adult-specific conditional deletion and thus reflects the maintenance, rather than the specification, of VCS cell fate. Accordingly, we have revised the text to explicitly state that *Tbx3* and *Tbx5* are critical for maintaining VCS identity in adult mice, and that their loss leads to a shift toward a working myocardial fate.

Minor:(1) There is no consistency in the way the quantitative data is shown in graphs. There are some graphs showing only bars, other dot plots, and other a combination of both. The authors must homogenise the representation of quantitative data showing the different data points in dot plots and not in bar graphs.

We have standardized the quantitative data presentation across all figures, by including individual data points in bar graphs, ensuring enhanced transparency and clarity.

(2) Figure 3: The labels defining the genotypes corresponding to the different histological sections of adult hearts (Panels G-K) are missing. Panels J and K are not referenced in the text.

We thank Reviewer #3 for highlighting these omissions. We have added the genotype labels to the histological sections in Panels G-K of Figure 3 to ensure clarity. Furthermore, we have now referenced Panels J and K in the results and in the supplementary material (please look below for references).

“Histological examination of all four-chambers demonstrated no discernible differences between VCS-specific *Tbx3:Tbx5* double-knockout (*Tbx3fl/fl;Tbx5fl/fl;R26EYFP/+; MinKCreERT2/+*) and control (*Tbx3+/+;Tbx5+/+;R26EYFP/+; MinKCreERT2/+*) mice, nor between . the double-knockout (*Tbx3fl/fl;Tbx5fl/fl;R26EYFP/+; MinKCreERT2/+*) and single-knockout models for either *Tbx3* (*Tbx3fl/fl;Tbx5+/+;R26EYFP/+; MinKCreERT2/+*) or *Tbx5* (*Tbx3+/+;Tbx5fl/fl;R26EYFP/+; MinKCreERT2/+*).Ventricular muscle appeared normal without hypertrophy or myofibrillar disarray and no fibrosis was present (Figure 3G, 3I, 3J, and 3K, respectively).”

“Additionally, we confirmed the absence of histological and structural abnormalities in these mice, aligning with previous findings (Figures 3A, 3F versus 3B, and 3K versus 3G, respectively)(1, 11).”

(3) Typo: Supplementary Figure 6. *Tbx3:Tbx3* double-conditional knockout: it should say *Tbx5:Tbx3* double-conditional knockout.

We thank Reviewer #3 for pointing out the typo. It has been corrected to: “Supplementary Figure 6. *Tbx3:Tbx5* double-conditional knockout mice exhibit QRS prolongation”.